# HEMM: Holistic Evaluation of Multimodal Foundation Models

**Paul Pu Liang**[*], **Akshay Goindani**[*], **Talha Chafekar, Leena Mathur, Haofei Yu,**
**Ruslan Salakhutdinov, Louis-Philippe Morency**
Machine Learning Department and Language Technologies Institute
Carnegie Mellon University
`https://github.com/pliang279/HEMM`

## Abstract

Multimodal foundation models that can holistically process text alongside images, video, audio, and other sensory modalities are increasingly used in a variety of real-world applications. However, it is challenging to characterize and study progress in multimodal foundation models, given the range of possible modeling decisions, tasks, and domains. In this paper, we introduce Holistic Evaluation of Multimodal Models (HEMM) to systematically evaluate the capabilities of multimodal foundation models across a set of 3 dimensions: basic skills, information flow, and real-world use cases. *Basic multimodal skills* are internal abilities required to solve problems, such as learning interactions across modalities, fine-grained alignment, multi-step reasoning, and the ability to handle external knowledge. *Information flow* studies how multimodal content changes during a task through querying, translation, editing, and fusion. *Use cases* span domain-specific challenges introduced in real-world multimedia, affective computing, natural sciences, healthcare, and human-computer interaction applications. Through comprehensive experiments across the 30 tasks in HEMM, we (1) identify key *dataset dimensions* (e.g., basic skills, information flows, and use cases) that pose challenges to today's models, and (2) distill performance trends regarding how different *modeling dimensions* (e.g., scale, pre-training data, multimodal alignment, pre-training, and instruction tuning objectives) influence performance. Our conclusions regarding challenging multimodal interactions, use cases, and tasks requiring reasoning and external knowledge, the benefits of data and model scale, and the impacts of instruction tuning yield actionable insights for future work in multimodal foundation models.

## 1 Introduction

Building upon rapid progress in large-scale language and vision pretraining [24, 69, 106], the new generation of multimodal foundation models is increasing adept at learning interactions between modalities [83], enables both static prediction and dynamic interaction [55], and even shows emergent properties never seen before in pretraining corpora [60]. Previous standards for benchmarking multimodal models based on collections of modality and task-specific datasets [8, 57, 29, 66] are increasingly insufficient in light of these general capabilities. In order to study fundamental questions regarding *why* multimodal foundation models exhibit certain behaviors, *when* they perform well in the real world, and *which* modeling paradigms are most effective, there is a need for a holistic evaluation scheme beyond individual datasets or contexts.

To address this need, we contribute **Holistic Evaluation of Multimodal Models (HEMM)**, visualized in Figure 1. HEMM, as an evaluation framework, goes beyond conventional lists of datasets to emphasize holistic benchmarking at three levels. The first level benchmarks *basic multimodal skills*: fundamental internal abilities required to address multimodal problems, such as interactions between

---

[*]These authors contributed equally to this work

38th Conference on Neural Information Processing Systems (NeurIPS 2024) Track on Datasets and Benchmarks.

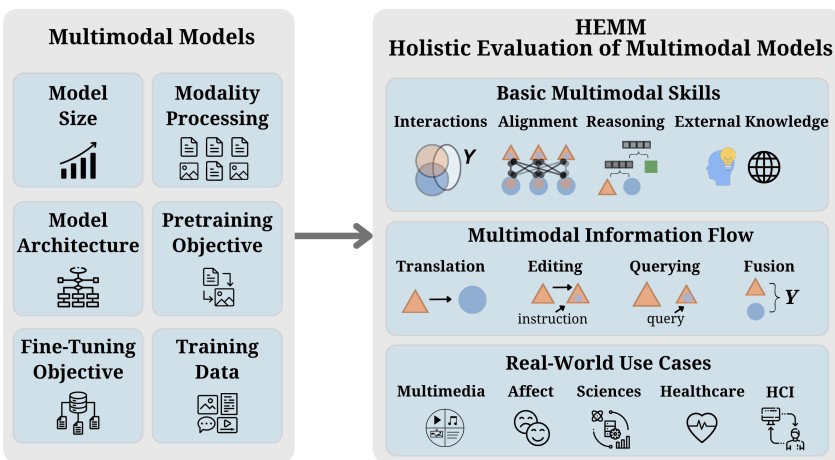

Figure 1: HEMM is an evaluation framework that characterizes multimodal models along several dimensions (size, architecture, pretraining objective, fine-tuning objective, training data) and emphasizes holistic benchmarking of these models at three disentangled levels: basic skills, information flow, and use cases.

redundant, unique, and synergistic features [26, 68], alignment of fine-grained and coarse-grained information [104], reasoning across compositional features [115], and integration of external knowledge [90]. The second level benchmarks *information flow*: how multimodal information transforms during tasks such as querying [98], translation [109], editing [108], and fusion [60]. The third level benchmarks *multimodal use cases*: how models perform in real-world challenges across domains, including multimedia, affective computing, natural sciences, healthcare, and human-computer interaction (HCI). Together, these three levels taxonomize a wide spectrum of 30 image-text datasets, enabling HEMM to serve as a holistic framework to evaluate multimodal models.

To aid in HEMM evaluation, we also present a new categorization of models spanning key *modeling decisions*, such as model size and modality processing (e.g., interleaved inputs), and *training decisions*, such as pretraining and fine-tuning objectives. We (1) identify key *dataset dimensions* (e.g., basic skills, information flows, and use cases) that pose challenges to today's models, and (2) distill performance trends regarding how different *modeling and training decisions* (e.g., scale, pre-training data, multimodal alignment, pre-training, and instruction tuning objectives) influence downstream task performance. Our analysis yields tangible directions for future work, including challenging multimodal skills, tasks, and use cases, impacts of diversity and scale, and guidelines on modeling architectures and training objectives. HEMM is publicly available at `anon`, and encourages community involvement in its expansion of datasets, annotations, models, and evaluation metrics.

## 2 Key Benchmarking Principles and Datasets in HEMM

HEMM includes 30 datasets summarized in Table 1. These datasets require different multimodal skills to solve, display different types of multimodal information flow, and belong to different real-world use cases with domain-specific challenges.

### 2.1 Basic multimodal skills

Multimodal skills are internal abilities required to solve multimodal tasks, such as learning interactions across modalities, fine-grained alignment, multi-step reasoning, and using external knowledge.

**Multimodal interactions** study how modality information is integrated for a multimodal task [69, 77, 52, 9], which can be *redundant*: shared between modalities, such as smiling while telling a humorous joke [43, 89], *unique*: present in only one of the modalities [35, 54], and *synergistic*: emergence of new information from both modalities, such as conveying sarcasm through conflicting verbal and nonverbal cues [15, 68]. Datasets with high referential information between modalities test for redundancy, such as in VQA, and translation on NoCAPS. Tasks with uniqueness or synergy include understanding movie posters (MM-IMDB), memes (MEMECAP), figurative language (IRFL), facial expressions (FER-2013), and cartoons (NEW YORKER CARTOON).

Table 1: HEMM includes a comprehensive suite of 30 datasets to benchmark multimodal foundation models. We categorize each dataset based on the *basic multimodal skills* needed to solve them – the type of multimodal interaction, granularity of multimodal alignment, level of reasoning, and need for external knowledge, how *information flows* between modalities, and the real-world *use cases* they impact.

| Dataset | # Samples | Interactions | Fine-grained | Reasoning | Knowledge | Info. Flow | Use case |
|---|---|---|---|---|---|---|---|
| VQA [4] | 614K | Redundancy | Yes | Less | No | Querying | Multimedia |
| VISUAL GENOME [50] | 1.7M | Redundancy | Yes | Less | No | Querying | Multimedia |
| VCR [123] | 290K | Redundancy | Yes | Less | No | Fusion | Multimedia |
| OK-VQA [76] | 14K | Redundancy | Yes | Less | Yes | Querying | Multimedia |
| GQA [42] | 22M | Redundancy | Yes | Less | No | Querying | Multimedia |
| NoCAPS [2] | 15K | Redundancy | No | Less | No | Translation | Multimedia |
| FLICKR30K [119] | 30K | Redundancy | No | Less | No | Translation | Multimedia |
| WINOGROUND [98] | 1.6K | Redundancy | Yes | Less | No | Querying | Multimedia |
| NLVR [93] | 92K | Redundancy | Yes | Less | No | Querying | Multimedia |
| NLVR2 [94] | 107K | Redundancy | No | Less | No | Querying | Multimedia |
| IRFL [117] | 3.9K | Synergy | No | More | No | Fusion | Multimedia |
| MM-IMDB [5] | 25K | Synergy | No | Less | No | Fusion | Multimedia |
| MAGIC BRUSH [124] | 10K | Synergy | Yes | Less | No | Editing | Multimedia |
| LNCOCO [87] | 8.5K | Uniqueness | Yes | Less | Yes | Translation | Multimedia |
| NY CARTOON [37] | 364 | Synergy | No | More | Yes | Fusion | Affect |
| HATEFUL MEMES [46] | 10K | Synergy | No | More | Yes | Fusion | Affect |
| MEMECAP [43] | 560 | Synergy | No | More | Yes | Fusion | Affect |
| MEMOTION [89] | 10K | Synergy | No | More | Yes | Fusion | Affect |
| FER-2013 [32] | 30K | Uniqueness | No | Less | No | Querying | Affect |
| SCIENCEQA [75] | 21K | Synergy | No | Less | Yes | Fusion | Science |
| RESISC45 [18] | 31K | Uniqueness | No | Less | No | Querying | Science |
| UCMERCED LAND USE [114] | 2K | Uniqueness | No | Less | No | Querying | Science |
| INATURALIST [102] | 675K | Uniqueness | Yes | Less | Yes | Querying | Science |
| DECIMER [13] | 5K | Uniqueness | No | More | Yes | Translation | Science |
| PATHVQA [35] | 33K | Redundancy | Yes | Less | Yes | Querying | Healthcare |
| VQARAD [53] | 3.5K | Redundancy | Yes | More | Yes | Querying | Healthcare |
| OPENPATH [41] | 218K | Redundancy | Yes | More | Yes | Querying | Healthcare |
| SLAKE [72] | 13K | Redundancy | Yes | More | Yes | Querying | Healthcare |
| ENRICO [58] | 1.4K | Uniqueness | No | Less | No | Querying | HCI |
| SCREEN2WORDS [103] | 112K | Uniqueness | No | Less | No | Translation | HCI |

**Granularity of multimodal alignment** involves identifying alignment across elements in different modalities. For example, answering a question might require a model to perform fine-grained alignment to reference one specific object out of many possible objects in an image. Tasks that explicitly test for fine-grained alignment include localized reasoning on VISUAL GENOME, WINOGROUND, while tasks that emphasize coarse-grained alignment (e.g., making a prediction relevant to a whole image) include interpreting cartoon images [37], movie posters [5], and memes [46, 89, 43].

**Reasoning and external knowledge** involve the combination of local pieces of information to form increasingly rich and complex multimodal representations. For example, being able to perform multi-hop inference from Wikipedia text and images [76] or solving science questions given visual diagrams and executing multiple logical steps [75]. Tasks like WINOGROUND explicitly test for reasoning and tasks like OK-VQA are designed to assess external knowledge.

## 2.2 Multimodal information flow

Multimodal information flow studies how information transforms across tasks, including cross-modal translation, editing, querying, and fusion.

**Cross-modal translation** exploits shared information by mapping data in one modality to another. Examples include translating from text to image for image generation (e.g., LNCOCO) and translating from image to text for image captioning (e.g., NoCAPS, SCREEN2WORDS).

**Cross-modal editing** involves semantically editing data in one modality according to another modality (e.g., given an image, following a natural language instruction to "change the background from day to night"). The model takes in the original image (with potentially more reference images), along with a task description specifying the edit, and outputs the edited image. We use the MAGIC BRUSH dataset to test cross-modal editing.

**Cross-modal querying** involves a model's ability to answer natural language questions that query specific information about an input. The model takes in the original image, a text description, the

Table 2: Models used in HEMM, ranked from small to large, and categorized by #Param (model size), Data Size (pretraining data size), Data Diversity (pretraining data diversity), Training Type (end-to-end training or frozen alignment), INST (instruction tuning), Modality Proc (interleaved or separate modality inputs).

| Model | #Param | Data Size | Data Diversity | Training Type | INST | Modality Proc |
|---|---|---|---|---|---|---|
| KOSMOS-2 [85] | 1.6B | 90M | Yes | End-to-end | Yes | interleaved |
| OPENFLAMINGO [6] | 3.2B | 180M | No | Modular Fine-tune | No | interleaved |
| INSTRUCT-BLIP [22] | 4.0B | 244M | Yes | Modular Fine-tune | Yes | separate |
| LLAMA-ADAPTER [30] | 7.0B | 567K | No | Modular Fine-tune | Yes | separate |
| MPLUG-OWL [116] | 7.2B | - | Yes | Modular Fine-tune | Yes | separate |
| FUYU-8B [10] | 9.3B | - | Yes | End-to-end | No | interleaved |
| BLIP-2 [61] | 12.1B | 244M | No | Modular Fine-tune | No | separate |
| MINI-GPT-4 [128] | 13.0B | 5M | No | Modular Fine-tune | Yes | separate |
| EMU [95] | 14.0B | 82M | Yes | End-to-end | No | interleaved |
| GEMINI | - | - | Yes | - | Yes | interleaved |
| GPT-4V | - | - | Yes | - | Yes | - |

query, and must output the desired answer (typically in natural language). Querying can be done for visual scenes (GQA), environmental indicators (RESISC45), and medical data (VQARAD).

**Multimodal fusion** aims to learn interactions to combine information from different modalities, such as classifying diseases given x-ray images and medical tests, or detecting humor from cartoon images and captions. Multimodal fusion takes in the image, text, and a description of the task, and then outputs a prediction, which can include affective states like humor in NEW YORKER CARTOON, hate speech detection in HATEFUL MEMES, or in science problems (SCIENCEQA).

### 2.3 Real-world Use Cases

Each use case is drawn from a real-world application with their own specific challenges.

**Multimedia** includes efficient search, retrieval, indexing, and generation of digital content. Multimedia tasks in HEMM include question answering about images and videos (VQA, VCR), multimedia captioning (FLICKR30K, NOCAPS), compositional visual reasoning (WINOGROUND, NLVR), understanding cartoons, movie posters (MM-IMDB), memes (MEMECAP and MEMOTION), and figurative language (IRFL), and editing images (MAGIC BRUSH).

**Affective computing** aims to perceive human affective states (emotions, sentiment, personalities, humor, sarcasm, social interactions) [86], and is important for building emotionally and socially-intelligent AI [56, 78] and human-AI interaction [55]. HEMM includes NEW YORKER CARTOON (cartoon images and captions), HATEFUL MEMES (hateful content in memes), FER-2013 for facial expressions, MEMECAP for meme captioning, and MEMOTION for emotions in memes.

**Natural sciences** aims to deepen our knowledge of physical, chemical, biological, and environmental sciences. These can involve satellite images, chemical bonds, land and agriculture use, wildlife, and specific scientific terminologye [101]. Tasks in HEMM include SCIENCEQA testing different science topics and RESISC45 for land scene classification.

**Healthcare** involves integrating multimodal signals such as lab tests, imaging reports, and doctor-patient interactions to help doctors interpret high-dimensional data and assist them in diagnosis [48, 51]. We include processing text reports and medical images in the form of PATHVQA for pathology, VQARAD for radiology images, and SLAKE for medical visual question answering.

**HCI** involves user design, usability, user experience, and other challenges related to humans interacting with computers [81]. HCI tasks can involve visual information such as screen layouts, user actions, and feedback mechanisms. HCI tasks in HEMM include ENRICO for classifying mobile UI designs and SCREEN2WORDS for UI screen content summarization.

## 3 Key Modeling Principles and Models in HEMM

Table 2 summarizes the 11 models we evaluate in HEMM, which span different numbers of parameters, model architectures, training datasets, pretraining objectives, and fine-tuning objectives.

### 3.1 Modeling decisions

**Model parameters** Parameters can vary greatly across different multimodal models, from 100M params to approximately 1000B params. We consider models with total number of parameters less

Table 3: Performance on different dataset dimensions, as measured via the mean BARTscore on each dataset across all 11 tested multimodal models.

| Dimension | Category | Perf (↑) |
|---|---|---|
| Real-world use case | Multimedia | **31.30** |
| | Affect | 30.35 |
| | Health | 20.24 |
| | Science | 19.83 |
| | HCI | 15.70 |
| Multimodal interaction | Redundancy | 29.04 |
| | Uniqueness | 19.60 |
| | Synergy | **33.73** |
| Reasoning | More Reasoning | 27.50 |
| | Less Reasoning | 26.84 |
| Granularity | Fine-grained | 26.52 |
| | Coarse-grained | 27.52 |
| Knowledge | External | 23.51 |
| | None | **29.62** |
| Information flow | Querying | 25.88 |
| | Translation | 18.97 |
| | Fusion | **33.77** |

Table 4: Performance on different modeling decisions, as measured via the mean BARTscore for each model across all 30 tested multimodal datasets.

| Dimension | Category | Perf (↑) |
|---|---|---|
| Modeling decisions | | |
| Modality processing | Interleaved | 22.94 |
| | Separate | **28.58** |
| Model size | Small | 23.34 |
| | Medium | 23.87 |
| | Large | **42.33** |
| Training decisions | | |
| Training type | Modular | **24.92** |
| | End-to-end | 21.26 |
| Size of training data | Small | 16.80 |
| | Medium | 30.10 |
| | Large | **31.77** |
| Diversity of training data | Non-diverse | 21.71 |
| | Diverse | **30.15** |
| Instruction tuning | No | 22.49 |
| | Yes | **29.71** |

than or equal to 4B (e.g., INSTRUCT-BLIP) as *small*, whereas those having more than 4B parameters (e.g., FUYU-8B) are considered *medium*. GPT-4V and GEMINI are considered *large*.

**Modality processing** Some multimodal models (e.g., FUYU-8B) support interleaved inputs like "`<dog_img> This is a very cute dog.<cat_img> This is a very cute cat.`", unlike models that only support separate image and text queries (e.g., BLIP-2, MINI-GPT-4).

## 3.2 Training Characteristics

**Training type** End-to-end training involves fine-tuning unimodal encoders, pretrained LLMs, and a multimodal model jointly, as seen in EMU, FUYU-8B, etc. Another category operates by freezing unimodal encoders and LLM, and then training only a mapping that aligns frozen image features with frozen LLM features. These trainable mappings include Q-former [22] (used in INSTRUCT-BLIP), linear layers [128, 92] (used in MINI-GPT-4), and attention blocks used in OPENFLAMINGO.

**Size of pre-training data** We consider the total size of pre-training data used for training, including instruction and supervised data. EMU has *small* data scale, with less than 100M training data points. FUYU-8B has *medium* data-scale, with more than 100M training data points. While GPT-4V and GEMINI do not release data sizes, we estimate their size to be much larger than other models and therefore are considered to have *large* data scale.

**Diversity of pre-training data** We consider the diversity of multimodal tasks used for training, including visual QA, visual conversations, and interleaved images and text. INSTRUCT-BLIP and EMU are pre-trained on diverse data, in contrast to LLAMA-ADAPTER, OPENFLAMINGO, etc., which only use image captioning data for training.

**Instruction tuning** By transforming supervised tasks into an 'instruction' format, instruction tuning has been shown to benefit performance and improve the controllability of LLMs. MINI-GPT-4 and INSTRUCT-BLIP include an instruction tuning stage, while models like BLIP-2 do not.

## 4 Experiments

In this section, we discuss extensive experiments conducted to holistically evaluate the performance of multimodal foundation models based on HEMM.

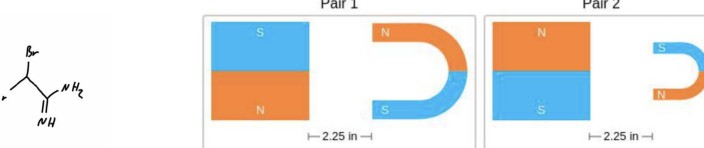

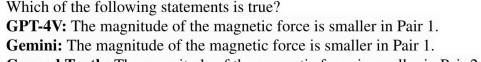

**Prompt:** Simplified molecular-input line-entry system (SMILES) notation of the given molecule:
**GPT-4V:** C(Br)(Br)C(=O)N
**Gemini:** BrC(Br)(NH2)NH2
**Ground Truth:** C(C(=N)N)(Br)Br

**Question:** Think about the magnetic force between the magnets in each pair. Which of the following statements is true?
**GPT-4V:** The magnitude of the magnetic force is smaller in Pair 1.
**Gemini:** The magnitude of the magnetic force is smaller in Pair 1.
**Ground Truth:** The magnitude of the magnetic force is smaller in Pair 2

**Prompt**: Given a Remote Sensing image scene, Classify the scene in following classes: ......., Answer:
**GPT-4V:** Runway
**Gemini:** Airport
**Ground Truth:** Runway

(a)                     (b)                     (c)

Figure 2: Responses of GPT-4V and GEMINI on samples from the science category. These failure cases show that the models lack domain knowledge and are unable to correctly translate the images of molecules to the SMILES notations (a). Example (b) shows that the models struggle on tasks requiring complex reasoning, failing to comprehend the relation between the force and the size of the magnets. In (c), all models except GPT-4V are unable to capture the fine-grained details and misclassify the image as an airport instead of a runway.

## 4.1 Experimental setup

**Individual metrics**    For all text generation tasks, we use the established natural language generation evaluation metric BARTScore [122], which was found to have the highest correlation with human judgement [122]. We compute BARTScore(r, c), where r is the reference and c is the candidate. It can be interpreted as the probability of generating the candidate sentence from the reference. For example, a model might caption an image with the following generated candidate: *A row of violins hanging on a wall.*. The reference (ground truth) of *A painting of 5 cello's with a green background* would be used to compute the BARTScore with respect to c.

**Aggregating metrics**    To aggregate scores across multiple tasks or models, we normalize scores using min-max scaling. Following Chang et al. [16], *min* represents the score of the worst multimodal model and *max* represents the identity score BARTScore(r, r), where r is the ground truth. Subsequently, these normalized scores in the 0 to 1 range can be interpreted as a percentage of model performance relative to the ground truth.

**Computation**    Since GPT-4V and GEMINI have query limits, we evaluate their performance on 100 random samples for each dataset (2800 total data points). For a fair comparison with other models, we present the results and findings below based on the performance of those 100 samples per dataset. In Appendix C we present the results of the other models on the full evaluation sets. We evaluate all the models on a single NVIDIA A100 80GB GPU with the inference time for a single image-text pair ranging from 0.1 seconds to 63.7 seconds. We report the average inference times for the models across all datasets and include additional details on the evaluation protocol in Appendix B.

## 4.2 Main results

We summarize our main results here and include full details in Appendix C. We first explain performance trends across the datasets in HEMM, before explaining performance differences across different multimodal foundation models and their design decisions.

### 4.2.1 Performance across dataset dimensions

**Overall comparisons**    We summarize overall trends in Figure 3 and Table 3. On average, models perform better on multimedia datasets, with IRFL (0.58), NLVR (0.50), and WINOGROUND (0.49) showing the highest scores. The lowest scores are for Healthcare, HCI, and Science use cases, such as on DECIMER (0.07), INATURALIST (0.08), ENRICO (0.12), PATHVQA (0.15), and MEMECAP (0.32). For predicting molecular structures on DECIMER, models are not able to generate correct chemical notations (in Simplified Molecular Input Line Entry System notation) and instead only generate names of individual atoms or compounds (see Figure 2). Other challenging datasets include INATURALIST due to fine-grained visual differences between 5000 species of plants and animals, and healthcare datasets that require intricate analysis of pathology images to identify organs, tissues, and anomalies (see Figure 8). Datasets related to memes were also challenging (0.32 and 0.38 on MEMECAP [43] and MEMOTION [89]), requiring knowledge about current events, pop culture, and metaphors beyond literal meanings.

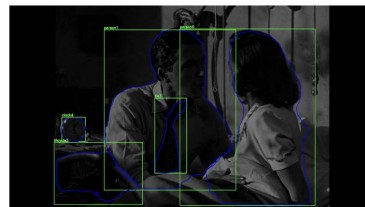 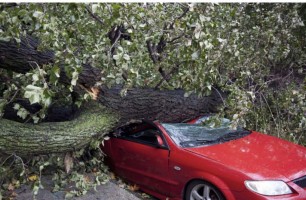 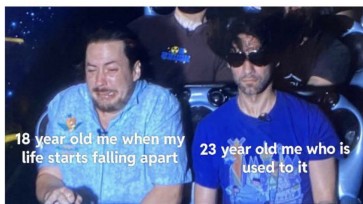

**Question:** Is person1 about to kiss person0? Choose from the below choices: No he won' t do that; No he is not going to kiss her; Yes, he is making a face for one; No, he is intruding. Answer:
**GPT-4V:** No, he is not going to kiss her.
**Gemini:** No, he is intruding.
**InstructBLIP:** Yes, he is making a face for one.
**Ground Truth:** Yes, he is making a face for one.

**Question:** Given an image and a text. Answer yes if the text matches the image and no if the text does not match the image. Text: a car smashed into a tree. Answer:
**GPT-4V:** Yes
**Gemini:** Yes
**InstructBLIP:** Yes
**Ground Truth:** No

**Question:** This is a meme with the title "The coaster of life man." The image description is "Two men in blue shirts on a roller coaster one with a pair of sunglasses on and the other is cringing.." What is the meme poster trying to convey? Answer:
**GPT-4V:** It conveys a humorous take on growing acclimatized to life's turbulent moments over time.
**Gemini:** The meme poster is trying to convey that life is a roller coaster. Sometimes it's up, sometimes it's down. But no matter what, you have to keep going.
**Ground Truth:** Meme poster is trying to convey that the older you get, the more easily you handle the very difficult things in life.

(a)                                  (b)                                  (c)

Figure 4: Tasks requiring commonsense and compositional reasoning are challenging. In (a), GPT-4V and GEMINI are unable to employ social commonsense to analyze the relationships between the two people. Example (b) demonstrates the models' difficulty in composing information from both modalities, leading to their failure to comprehend the scenario where *a tree smashed into the car* (not a car smashed into the tree). In (c), all models except GPT-4V fail to grasp the visual metaphors and the juxtaposition of the two scenarios.

**Multimodal skills 1: Interactions** The average scores for redundant, unique, and synergistic interactions are 0.29, 0.20, and 0.33. One reason for lower uniqueness scores is the presence of highly challenging visual datasets like DECIMER and ENRICO. On average, the easiest tasks in redundancy are NLVR (0.50) and WINOGROUND (0.49). The hardest datasets in uniqueness are INATURALIST (0.08) and DECIMER (0.07), and in synergy are MEMECAP (0.14) and MEMOTION (0.21).

**Multimodal skills 2: Granularity** We do not find that fine-grained datasets are significantly harder than those with coarse-grained alignment. Tasks requiring fine-grained alignment between image and text like GQA and WINOGROUND achieve a score of 0.26, while those only needing coarse-grained alignment (e.g., ENRICO, SCIENCEQA) are still quite challenging (score: 0.27).

**Multimodal skills 3: Reasoning** We do not find a significant difference between the perfor-

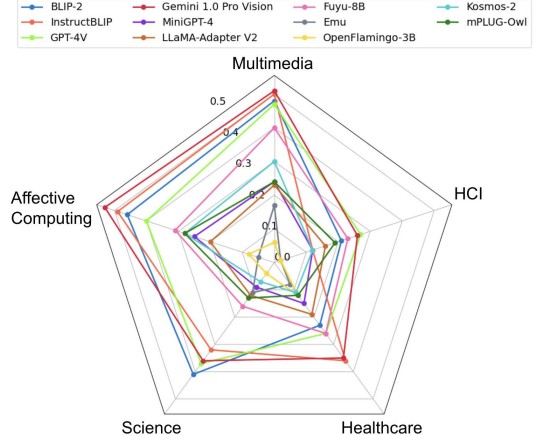

Figure 3: Average scores are higher for multimedia datasets as compared to other use cases, and lowest for healthcare, HCI, and science. The models struggle on INATURALIST, DECIMER, ENRICO, PATHVQA, and MEMECAP which require external knowledge, fine-grained alignment, and complex reasoning.

mance of the models on tasks requiring more (average score = 0.275) or less reasoning (average score = 0.268). The most challenging datasets requiring less reasoning include INATURALIST (0.08) and ENRICO (0.12) due to challenges in fine-grained visual perception and external knowledge, while there are also several challenging datasets requiring more complex reasoning like VCR (0.34) and MEMECAP (0.14), where the models encounter difficulties with samples requiring commonsense and compositional reasoning (See Figure 4 for examples).

**Multimodal skills 4: External knowledge** The average performance on tasks requiring external knowledge is 0.23, compared to 0.30 for those not requiring external knowledge. For example, INSTRUCT-BLIP performs well on WINOGROUND and VCR that do not require external knowledge but struggles more on knowledge-intensive tasks e.g., INATURALIST, which requires knowledge

about characteristics of a vast number of species, and SLAKE, where medical knowledge is needed to identify the abnormalities in organs.

**Multimodal Skills 5: Information flow**    Translation has the lowest average score amongst all types of information flow (0.19), whereas the average scores on querying and fusion are 0.26 and 0.33 respectively. The low performance on translation is due to the presence of challenging datasets like DECIMER and SCREEN2WORDS requiring mapping images of chemicals and screenshots into text. Although the average score for fusion is high, the performance on some datasets is still quite low, such as INSTRUCT-BLIP achieving a score of only 0.04 on MEMECAP and 0.15 on MM-IMDB.

#### 4.2.2   Performance across modeling dimensions

We now compare different modeling decisions and training objectives in Table 4.

**Overall comparisons across models**    GEMINI [97] (0.44), INSTRUCT-BLIP [22] (0.41), BLIP-2 [62] (0.41), and GPT-4V [1] (0.40) achieve the best average performance across all tasks. The low scores of GPT-4V as compared to GEMINI and INSTRUCT-BLIP are due to its generation of keywords like "Indeterminate", "Uncertain", and "Unknown" on datasets like VQA and GQA, perhaps due to its alignment process. Further, on some datasets related to Memes (e.g., HATEFUL MEMES) and Health (e.g., SLAKE), GPT-4V refrains from answering the questions and instead generates a response saying *Cannot assist with the request*. OPENFLAMINGO [6] (0.06), EMU [95] (0.11) have the lowest average scores. From their generations, we find that these models struggle to follow the instructions for challenging datasets like DECIMER and ENRICO, and generate hallucinated responses. Moreover, with relatively easier datasets such as FLICKR30K, the captions produced by EMU and OPENFLAMINGO tend to fixate on specific objects rather than providing a comprehensive description of the scene, often leading to instances of hallucination related to these objects. As a result, these models rank lowest on many datasets, receiving a normalized score of 0.

**Model scale**    We find that the performance of larger models (both total and trainable parameters) is significantly better than the models with a medium or small number of parameters (Figure 5). When grouped based on the total number of parameters, the average scores achieved by large, medium, and small models are 0.42, 0.24, and 0.23 respectively. The difference between the performance of large and medium models is significant (p-value for paired t-Test < 0.001). In particular, large models showed the most improvement on MM-IMDB, MEMECAP, and HATEFUL MEMES datasets, which fall into the category of tasks requiring synergistic interactions. On average, the large models perform the

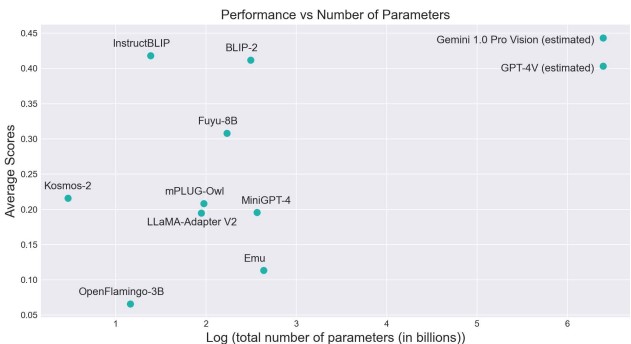

Figure 5: On average, large models are better than small and medium models (p-values < 0.001). INSTRUCT-BLIP and BLIP-2 are outliers - despite having fewer params, they achieve relatively high performance, even close to GPT-4V and GEMINI.

best on synergistic tasks with a score of 0.53 compared to 0.30 for medium and 0.23 for small models. For instance, on the MM-IMDB dataset, we observed significant gains in performance when increasing model size: from 0.15 for INSTRUCT-BLIP (small) to 0.36 for BLIP-2 (medium) and 0.48 for GEMINI (large).

**Pretraining data scale**    Average scores of the models in *large* and *medium* data size categories are 0.31 and 0.30 respectively, whereas models with *small* pretraining data achieve a significantly lower score of 0.17. We also find that for all datasets, the average score of models with *medium* pretraining data is higher than the models with *small* pretraining data. For instance, on the WINOGROUND dataset which requires fine-grained alignment between the modalities, the maximum scores achieved by the models with *medium* and *small* pretraining data are 0.45 and 0.80. We also find a significant gap between the maximum scores achieved by the models in the *medium* (maximum score - 0.18) and *small* categories (maximum score - 0.70), on the NLVR2 dataset for visual reasoning.

**Diversity of pre-training data**    On average, models trained on diverse datasets perform better (score: 0.30) than models trained only on image captioning datasets (score: 0.21). Diverse training

data allows the models to share learned knowledge and generalize across different tasks. For example, models pretrained with diverse datasets perform significantly better on the knowledge-intensive INATURALIST task, such as BLIP-2 (non-diverse) scoring 0.08 and GEMINI scoring 0.24. For the MEMECAP dataset which requires external knowledge and complex reasoning, we observe that BLIP-2 (non-diverse) scores 0.06 and MPLUG-OWL (diverse) scores 0.21.

**Instruction tuning vs supervised fine-tuning**  On average, instruction-tuned models (average score of 0.30) performed better than the models trained using only supervised fine-tuning (average score of 0.22). The top 3 tasks with the largest performance gap between instruction-tuned and non-instruction-tuned models are DECIMER, MEMECAP, and SCREEN2WORDS, with improvements of 0.15, 0.09, and 0.09 respectively. We also observe that translation tasks (image-to-text) (e.g., FLICKR30K, NOCAPS) benefit from instruction tuning, where the models generate more accurate and detailed captions after human instruction.

### 4.3   Human evaluation

To assess how well HEMM aligns with human preferences, we performed human preference-based evaluation, following Chiang et al. [19], where annotators are shown the outputs of two different models for the same inputs and choose the better output or a tie option. Across 1000 pairwise comparisons by 5 annotators, the pairwise rankings are used to calculate each model's average win rate and Elo rating (see Appendix B.5 for calculation details).

The models ranked by Elo ratings are GEMINI (1074), GPT-4V (1057), BLIP-2 (1033), and INSTRUCT-BLIP (1032) (see Table 5). The top 4 models based on the Elo Rating are the same as the top 4 models ranked by BARTScore. Elo Rating of GPT-4V is better than BLIP-2 and INSTRUCT-BLIP. However, the average BARTScore for GPT-4V (0.40) is lower than INSTRUCT-BLIP (0.42) and BLIP-2 (0.41). We also find Elo Rating of bottom two models to be consistent with BARTScore rankings - EMU (0.11) and OPENFLAMINGO (0.06).

Table 5: Average win rate and Elo Rating of 11 models calculated based on the human evaluation of 1000 pair-wise comparisons of model responses. Elo rating is reported as the median over 1000 runs with shuffled battle sequences and an initial rating of 1000 for each model. Top 4 and bottom 2 models identified by Elo Rating are consistent with those found by Average BARTScore.

| Model | Avg. Win Rate | Elo Rating | Avg. BARTScore |
|---|---|---|---|
| GEMINI | **0.73** | **1074** | **0.44** |
| GPT-4V | 0.68 | 1057 | 0.40 |
| BLIP-2 | 0.52 | 1033 | 0.41 |
| INSTRUCT-BLIP | 0.60 | 1032 | 0.42 |
| MPLUG-OWL | 0.45 | 1010 | 0.21 |
| LLAMA-ADAPTER | 0.45 | 1008 | 0.19 |
| FUYU-8B | 0.42 | 992 | 0.31 |
| MINI-GPT-4 | 0.38 | 990 | 0.20 |
| KOSMOS-2 | 0.39 | 968 | 0.22 |
| EMU | 0.20 | 924 | 0.11 |
| OPENFLAMINGO | 0.17 | 911 | 0.06 |

## 5   Related Work

**Multimodal machine learning** brings unique challenges for ML research due to the heterogeneity between modalities and the interconnections found between them [69]. It has inspired many theoretical studies in data heterogeneity and interactions [25], as well as diverse applications in multimedia [44, 14, 88], affective computing [86], robotics [47], finance [39], HCI [25, 82], education [12] and healthcare [80, 110].

**Evaluation frameworks for multimodal models** have significantly shaped the multimodal research landscape, through holistic [57, 66] and domain-specific benchmarks [31, 28]. Recent benchmarks have focused on testing the capabilities of multimodal foundation models, such as MME [29], MMBench [73], LVLM-ehub [111], SEED-Bench [59], Touchstone [7], Mm-vet [121], ReForm-Eval [65], VisIT-Bench [11], FLAVA [45]. Other benchmarks focus on evaluating hallucination [21] and applications in medicine [113] and autonomous driving [107]. These benchmarks contain many tasks, but without the systematic taxonomy and comprehensiveness that HEMM provides.

**Multimodal foundation models** are promising foundations for the future of AI, with impressive reasoning [75], interactive dialogue [49], and few-shot generalization abilities [100]. These models can be pre-trained (typically with image-text self-supervised learning) and fine-tuned for downstream tasks [63, 74, 91, 67], or based on adapting language models with vision to enable text generation conditioned on images [61, 105]. Cross-modal transformer architectures have emerged as a popular backbone due to their suitability for both language and image data [17, 99]. Additionally, composable

models [96] and mixtures of experts [120] can be used to further generate combinations of output modalities.

**Adapting language models for multimodality** is another promising approach where frozen models are aligned on both vision and language to generate text from multimodal inputs [128, 62, 118, 109]. These approaches typically use parameter-efficient modules like LLaMA-Adapter V2 [30] and MAGMA [27] for efficient finetuning. Vision-language instruction tuning has also emerged as a useful technique, as it allows the models to better follow human instructions [112, 128]. Our goal is to make HEMM the most comprehensive benchmark to study the current and future generation of multimodal foundation models, and for the community to continuously contribute to its expansion.

# 6 Conclusion

Holistic Evaluation of Multimodal Models (HEMM) is a framework for benchmarking multimodal foundation models. Through a new taxonomy of multimodal skills, information flow, and real-world use cases, HEMM enables comprehensive analysis of multimodal models. HEMM is publicly available, will be regularly updated, and encourages community involvement in its expansion.

**Limitations and social impact** The evaluation of multimodal models is done only on a subset of all possible skills, information, and use cases in the world. Future work can improve the categorization of datasets into skills, information, and use cases, and discover new dimensions that pose challenges to multimodal models. Such evaluation is critical to ensure that models are sufficiently robust when deployed in real-world scenarios, to prevent unexpected and unintended consequences. Future work should also add new metrics to HEMM measuring real-world societal concerns such as fairness, robustness, social biases, privacy, and efficiency of multimodal models.

# Acknowledgements

This material is based upon work partially supported by National Science Foundation awards 1722822 and 1750439, National Institutes of Health awards R01MH125740, R01MH132225, R01MH096951 and R21MH130767, and Meta. PPL is supported in part by a Siebel Scholarship and a Waibel Presidential Fellowship. RS is supported in part by ONR grant N000142312368 and DARPA FA87502321015. Any opinions, findings, conclusions, or recommendations expressed in this material are those of the author(s) and do not necessarily reflect the views of the sponsors, and no official endorsement should be inferred. We are also grateful to NVIDIA's GPU support.

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

# Appendix

## A  HEMM Details

### A.1  Individual dataset details

In this section we provide the details of the tasks and datasets chosen for the HEMM benchmark: we describe the split used to evaluate the models, any prepossessing applied to the samples, and their access restrictions and licenses.

1. **VQA** dataset consists of samples of an image and a corresponding free-form, open-ended question. To answer the questions, the models need to perform fine-grained recognition of objects and activities. Some of the samples require commonsense reasoning to correctly answer the questions. Most of the samples in the dataset have "yes" or "no" answers.

   **Split:** We evaluate on the real images validation set which comprises of a total of 244,302 questions.

   **Prompt used:** You are given an image and a question. Answer the question in a single word. Question: *<question>*

   **Access restrictions:** The dataset is available to download from https://visualqa.org/vqa_v1_download.html

   **Licenses:** The images in the dataset come with a CC BY 4.0 DEED license https://creativecommons.org/licenses/by/4.0/deed.en

   **Ethical considerations:** No personally identifiable information or offensive content present in the dataset.

2. **NoCaps** dataset is a large scale image captioning dataset. Training data for this dataset consists of Image-Caption pairs from COCO dataset [71] as well as images and labels from Open Images. Many objects seen in the test set have very few associated captions from the training set making it a robust benchmark for image captioning.

   **Split:** Evaluation is performed on the validation set which consists of 4500 images.

   **Prompt used:** You are given an image. This image might contain a lot of objects. You have to generate a caption for the image but the caption should just be a single sentence. Please do not generate more than one sentences. Caption:

   **Access restrictions:** The dataset is available to download from https://nocaps.org/download

   **Licenses:** Images belonging to the dataset are available under CC BY 2.0 https://creativecommons.org/licenses/by/2.0/deed.en license which permits to redistribute the data.

   **Ethical considerations:** No personally identifiable information or offensive content present in the dataset.

3. **Decimer** dataset is a hand-drawn molecule image dataset consisting of chemical structure as the images and their SMILES representation as the strings. This SMILES representation stands for 'Simplified Molecular Input Line Entry System', which depicts the three-dimensional structure of the chemical into a string of symbols. In order to solve this task, the model should have an understanding of structure of the chemical and how these structures are depicted in the given format.

   **Split:** The dataset consists of 5088 images over which evaluation has been performed.

   **Prompt used:** Simplified molecular-input line-entry system (SMILES) notation of the given molecule:

   **Access restrictions:** The dataset is available to download from https://zenodo.org/records/7617107

   **Licenses:** The dataset is available under Creative Commons Attribution 4.0 International License https://creativecommons.org/licenses/by/4.0/deed.en, which permits use and sharing of data.

**Ethical considerations:** No personally identifiable information or offensive content present in the dataset.

4. **MEMOTION** dataset was introduced in the 'Memotion Analysis' challenge. This task consisted of three different tasks: sentiment classification, humor classification, and the scale of semantic classes. In our evaluation, we focus on the scale of humor class which consists of 'funny', 'very funny', 'not funny', and 'hilarious'. Images in this dataset consists of memes from the internet, which have been annotated by humans for their class labels.

   **Splits:** A total of 6992 images were used.

   **Prompt used:** Question: Given the Meme and the following caption, Caption:*<caption>*. How funny is the meme? Choose from the following comma separated options: funny, very funny, not funny, hilarious.

   **Access restrictions:** The dataset is available to download from https://www.kaggle.com/datasets/williamscott701/memotion-dataset-7k

   **Licenses:** The dataset is available under Creative Commons Attribution 4.0 International License https://creativecommons.org/licenses/by/4.0/deed.en which allows sharing of data.

   **Ethical considerations:** No personally identifiable information is present in the data. Offensive content is present in the dataset in some meme images.

5. **SCIENCEQA** consists of multiple choice questions from different science topics consisting of natural science, social science, and language science. The model has to choose an answer from the given set of options for a question, by making sense of lecture and explanation which are optional for a question. Some questions do not consist of an image, however, we evaluate only on questions that have an image in the data point.

   **Split:** A total of 4.24k questions from the test set.

   **Prompt used:** You are given a question and few choices. There is context provided with the image which will help you to understand the image. To answer the question, you have been given lecture notes. You can use these lecture notes, image, and context to answer the question. There are some choices given to you which are comma-separated. You have to select which choice best answers the question. Generate choice as it is from the given choices. lecture: *<lecture>* question: *<question>* context: *<context>* choices: *<choices>* Answer:

   **Access restrictions:** The dataset is available to download from https://huggingface.co/datasets/derek-thomas/ScienceQA

   **Licenses:** Dataset is distributed under the CC BY-NC-SA (Attribution-NonCommercial-ShareAlike) https://creativecommons.org/licenses/by-nc-sa/4.0/deed.en which allows sharing data.

   **Ethical considerations:** No personally identifiable information or offensive content is present in the dataset.

6. **SLAKE** is a medical visual question-answering dataset that consists of image and question-answer pairs. Annotations have been done by experienced physicians and a medical knowledge base for medical visual question answering. The dataset consists of Yes/No type of questions as well as questions which could be answered with a single word.

   **Split:** We use the test set of this dataset which consists of 2070 questions.

   **Prompt used:** Answer the question in a single word, Question: *<question>*

   **Access restrictions:** The dataset is available to download from https://huggingface.co/datasets/BoKelvin/SLAKE

   **Licenses:** Images under this dataset are available in CC-BY-SA 4.0 license https://creativecommons.org/licenses/by-sa/4.0/deed.en which allows sharing data.

   **Ethical considerations:** No personally identifiable information or offensive content is present in the dataset.

7. **VISUAL GENOME** dataset is a visual question-answering dataset that grounds visual concepts to language. Visual Genome provides a formal representation of an image, as relationships between objects in the image are depicted with the help of a scene graph. WordNet [79] is used to canonicalize objects, attributes, and relationships in each image.

   **Split:** We use the splits available from https://homes.cs.washington.edu/ ranjay/visualgenome/data/dataset/question_answers.json.zip

   **Prompt used:** You are given an image and a question. Answer the question in a single word only. Question: *<question>*

   **Access restrictions:** The dataset is available to download from https://homes.cs.washington.edu/ ranjay/visualgenome/api.html

   **Licenses:** The data is available under Creative Commons Attribution 4.0 International License https://creativecommons.org/licenses/by/4.0/deed.en

   **Ethical considerations:** No personally identifiable information or offensive content is present in the dataset.

8. **PATHVQA** is a visual QA dataset based on pathology images, PathVQA consists of images taken from pathology textbooks and online digital libraries, with question-answer pairs generated from captions using a question generation pipeline. Each pathology image is coupled with a question-answer pair.

   **Split:** The test set consists of 6,012 questions.

   **Prompt used:** You are given a radiology image and a question. Answer the question in a single word. Question: *<question>*

   **Access restrictions:** The data is available to download from https://github.com/UCSD-AI4H/PathVQA

   **Licenses:** No licenses are available for this dataset.

   **Ethical considerations:** No personally identifiable information or offensive content is present in the dataset.

9. **UCMERCED LAND USE** is another dataset for land use classification which has 21 classes. Images from the USGS National Map Urban Area Imagery were extracted manually, which involves various urban areas around the country. We include all the possible classes in the prompt so the model can choose from them.

   **Split:** We evaluate on the validation split present in https://www.kaggle.com/datasets/apollo2506/landuse-scene-classification.

   **Prompt used:** Image is given to you. Classify if the image belongs to one of the following classes: mediumresidential, buildings, tenniscourt, denseresidential, baseballdiamond, intersection, harbor, parkinglot, river, overpass, mobilehomepark, runway, forest, beach, freeway, airplane, storagetanks, chaparral, golfcourse, sparseresidential, agricultural. Choose a class from the above classes.

   **Access restrictions:** The dataset is available to download from http://weegee.vision.ucmerced.edu/datasets/landuse.html or https://www.kaggle.com/datasets/apollo2506/landuse-scene-classification

   **Licenses:** No licenses are available for this dataset.

   **Ethical considerations:** No personally identifiable information or offensive content is present in the dataset.

10. **ENRICO** is a topic modeling dataset for mobile UI screens. It is an enhanced version of RICO dataset [23] where samples were ranked as a good or bad design example by two human annotators. UI classes in the dataset consist of interfaces such as calculator, camera, chat, news, profile, etc from which the model has to choose for a particular image.

    **Split:** We evaluate on the dataset provided in http://userinterfaces.aalto.fi/enrico/resources/screenshots.zip

**Prompt used:** Given a screenshot of the user interface of a mobile application. Choose the most appropriate design topic from the following comma-separated choices: bare, dialer, camera, chat, editor, form, gallery, list, login, maps, mediaplayer, menu, modal, news, other, profile, search, settings, terms, tutorial

**Access restrictions:** The dataset is available to download from https://github.com/luileito/enrico

**Licenses:** The dataset comes under MIT license https://github.com/luileito/enrico/blob/master/LICENSE

**Ethical considerations:** No personally identifiable information or offensive content is present in the dataset.

11. **MM-IMDB** is a genre prediction dataset that consists of an image of the poster of the movie along with the plot. Each movie can belong to multiple genre. This dataset was built with MovieLens 20M dataset [34] which consists of movie ratings. Using this, information such as genre, plot, year, and additional metadata were collected. For our evaluation, only poster image and plot is used for genre prediction.

    **Split:** We evaluate on the test split.

    **Prompt used:** Given the movie poster and the corresponding plot of the movie, choose the appropriate genres from the following comma-separated genres: drama, comedy, romance, thriller, crime, action, adventure, horror, documentry, mystery, sci-fi, fantasy, family, biography, war, history, music, animation, musical, western, sport, short, film-noir. Plot: <*plot*> Note that a movie can belong to more than one genres, provide all the suitable genres seperated by commas.

    **Access restrictions:** It is a public dataset free to download by the research community from http://lisi1.unal.edu.co/mmimdb/ and https://github.com/johnarevalo/gmu-mmimdb/

    **Licenses:** The dataset comes under MIT license https://github.com/johnarevalo/gmu-mmimdb/blob/master/LICENSE

    **Ethical considerations:** No personally identifiable information or offensive content is present in the dataset.

12. **VQARAD** is a visual question-answering dataset over radiology images. Images are taken from MedPix[*] an open radiology database. The dataset is constructed manually by clinical annotators consisting of medical students and senior radiologists. Ground truth answers for the questions are related to counting, color, abnormality, and presence of condition among others.

    **Split:** We evaluate on the test set present in https://huggingface.co/datasets/flaviagiammarino/vqa-rad/viewer/default/test which consists of 451 questions.

    **Prompt used:** You are given a radiology image and a question. Answer the question in a single word. Question:<*question*>

    **Access restrictions:** The dataset is available at https://huggingface.co/datasets/flaviagiammarino/vqa-rad/viewer

    **Licenses:** The dataset is available under Creative Commons Attribution 4.0 International License https://creativecommons.org/licenses/by/4.0/deed.en

    **Ethical considerations:** No personally identifiable information or offensive content is present in the dataset.

13. **FLICKR30K** is an image captioning dataset collected from Flickr[*] which extends [38] dataset with similar dataset collection and annotation guidelines.

    **Split:** We evaluate the dataset on the test split.

---

[*]https://medpix.nlm.nih.gov/home
[*]https://www.flickr.com/

**Prompt used:** A Picture of

**Access restrictions:** The dataset is available to download from https://www.kaggle.com/datasets/hsankesara/flickr-image-dataset

**Licenses:** The dataset is available under CC0: Public Domain License https://creativecommons.org/publicdomain/zero/1.0/deed.en

**Ethical considerations:** No personally identifiable information or offensive content is present in the dataset.

14. **FER-2013** is a classic dataset for facial expression recognition, where each image has to be classified into 7 labels. Images for this dataset were obtained from Google images, by searching them using Google Search API. OpenCV was used to get bounding boxes for faces in each of the images.

    **Split:** We evaluate on the test dataset present in https://www.kaggle.com/datasets/msambare/fer2013

    **Prompt used:** Given the photo of a face, determine the face expression, choose from the following choices: angry, disgust, fear, happy, neutral, sad, surprise. Answer in a single word.

    **Access restrictions:** The dataset is available to download from https://www.kaggle.com/datasets/msambare/fer2013

    **Licenses:** No license is provided with the dataset

    **Ethical considerations:** This dataset contains human faces collected through Google image search queries but does not contain any identifying information about user identities and backgrounds. No offensive content is present in the dataset.

15. **NY Cartoon** is collected from the weekly New Yorker magazine cartoon captioning contest [*], where readers are tasked to give a humorous caption for a cartoon image and the funniest captions are selected based on public votes. The dataset is formulated based on taking in the image and caption to predict how funny the pair is based on the normalized number of votes. Given an image and its caption, we ask the model if the caption is humorous or not. Each image has multiple caption choices with votes for the caption being not funny, somewhat funny, funny. We select the funniest caption to have a ground truth answer as 'yes' when prompted for evaluation. The next four funniest captions are selected to have ground truth answers as 'no' when prompted for evaluation.

    **Split:** We use the data available on https://github.com/nextml/caption-contest-data

    **Prompt used:** You are given a cartoon image and a caption. start the answer with yes if the caption is funny or No if the caption is not funny. Caption: *<caption>*

    **Access restrictions:** The dataset is available to download from https://github.com/nextml/caption-contest-data

    **Licenses:** No license is provided with the dataset.

    **Ethical considerations:** No personally identifiable information or offensive content is present in the dataset.

16. **OK-VQA** is a visual question-answering task that requires outside knowledge and reasoning to answer questions. Images for this dataset are taken from the COCO dataset[71] and MTurk [*] is used for labeling questions. A specific instruction is given to the workers to label questions that require knowledge outside the image. In this dataset, questions are of open-ended type.

    **Split:** We use the test set available here https://okvqa.allenai.org/download.html

    **Prompt used:** You are given an image and a question. Answer the question in a single word. Question: *<question>*

---

[*]https://www.newyorker.com/cartoons/contest
[*]https://www.mturk.com/

17. **MAGIC BRUSH** is an instruction-based image editing dataset consisting of manually annotated images consisting of single-turn and multi-turn instruction-guided editing. Images are sampled from MS COCO [71] dataset and are annotated using DALL-E 2 [*] with the help of crowdworkers from Amazon Mechanical Turk (AMT)[*]. For our evaluation, we follow a single-turn instruction editing.

    **Split:** We evaluate on the test set available from https://osu-nlp-group.github.io/MagicBrush/

    **Prompt used:** Edit the given image based on the provided instruction. Instruction: *<instruction>*

    

18. **MEMECAP** is a meme captioning dataset, whose images have been taken from the subreddit r/memes [*]. The captions for these images are generated in a two-round process by human annotators using Amazon Mechanical Turk. For our evaluation process, we provide the model with the image description and title of the meme and ask what the meme is trying to convey.

    **Split:** We evaluate on the test set from https://github.com/eujhwang/meme-cap/tree/main

    **Prompt used:** This is a meme with the title *<title>*. The image description is **. What is the meme poster trying to convey? Answer:

    

19. **HATEFUL MEMES** was a challenge hosted by Meta to classify if a meme image along with its text caption describes hateful intentions. Images were obtained from Getty images[*] annotated by a third-party annotation platform. Here, an image and text are provided to the model to ask if the image promotes hateful sentiments.

    **Splits:** We use the 'dev' split from https://www.kaggle.com/datasets/parthplc/facebook-hateful-meme-dataset/data

    **Prompt used:** You are given an image. In the image, the text phrase that you will be given and the image are innocuous when considered by themselves. The semantic content of the meme becomes mean only when the text phrase and image are considered together. Text phrase: *<text_phrase>* You have to judge if the combination of image and text is hateful or not. Always begin your answer with either 'yes' or 'no' with 'yes' indicating that the meme is hateful and 'no' if it is not hateful. Answer:

---

[*]https://openai.com/dall-e-2

[*]https://www.mturk.com/

[*]https://www.reddit.com/r/memes/

[*]https://www.gettyimages.in/

20. **INATURALIST** is an image classification dataset for 5000 wildlife species of plants and animals. Images and labels are sourced from iNaturalist website [*]. We evaluate the models by asking them to identify the species present in the given image. We do not provide it with possible classes as the dataset spans over a set of 5000 species.

    **Split:** We evaluate the model on the validation split provided in the 2021 edition of the dataset.

    **Prompt used:** The scientific species name of the species present in the image is:

    

21. **NLVR** consists of image-text pairs for visual reasoning. Images are created by generating objects and their properties randomly. These images are then given to a crowd worker to describe the image in a sentence.

    **Split:** Data is evaluated on the dev split from https://github.com/lil-lab/nlvr

    **Prompt used:** Given this image along with a question about the image, please answer the question with only the word 'true' or 'false'. Question: *<question>*

    

22. **NLVR2** extends NLVR to real-world photographs, and captions for these photographs. Images are retrieved using search queries from the ILSVRC2014 ImageNet challenge [*]. Crowdworkers are used to write the captions for the images. For this dataset, each data point has two images and a sentence that talks about the images. We concatenate the two images so that we pass a single image in the model.

    **Split:** Evaluation is performed on the dev split from https://github.com/lil-lab/nlvr/tree/master/nlvr2

    **Prompt used:** You are given an image and a related text, use the image as context and reply with true or false only Text: *<text>* Answer:

    

---

[*]https://www.inaturalist.org/
[*]https://www.image-net.org/challenges/LSVRC/2014/

**Ethical considerations:** No personally identifiable information or offensive content is present in the dataset.

23. **VCR** tests commonsense reasoning skills in question answering over images. Still images are extracted from movie clips, and annotations are crowdsourced using Amazon Mechanical Turk where each worker is provided an image along with detailed video captions to collect questions, answers, and rationales for an image

    **Split:** 'val' split is used from https://visualcommonsense.com/download/

    **Prompt used:** Question: *<question>* Choose from the below choices: *<choices>*

    **Access restrictions:** The dataset is available to download from https://visualcommonsense.com/download/

    **Licenses:** The dataset is provided in license as https://visualcommonsense.com/license/

    **Ethical considerations:** No personally identifiable information or offensive content is present in the dataset.

24. **WINOGROUND** is a dataset for visual linguistic compositional reasoning involving images from Getty Images and annotations given by four expert annotators. The original task consists of matching images and captions for a pair of two images and captions. We transform this task by creating a total of four data points for each pair by pairing each caption, with each image which leads to two correct and two wrong pairs per data point. We then ask the model to see if the caption matches the pair or not.

    **Split:** Test set from https://huggingface.co/datasets/facebook/winoground is used.

    **Prompt used:** You are given an image and a text. Answer yes if the text matches the image and no if the text does not match the image. Text: *<text>* Answer:

    **Access restrictions:** The dataset is downloaded from https://huggingface.co/datasets/facebook/winoground

    **Licenses:** Authors of the dataset have Getty image license https://www.gettyimages.in/eula.

    **Ethical considerations:** No personally identifiable information or offensive content is present in the dataset.

25. **RESISC45** is a land use dataset that involves land scene classification of images over 45 classes. The images for this dataset have been taken from Google Earth by experts in remote sensing image interpretation. We add all 45 classes to the prompt and let the model choose the class from the prompt itself.

    **Split**: We use the dataset from https://www.kaggle.com/datasets/happyyang/nwpu-data-set

    **Prompt used:** Image is given to you. Classify if the image belongs to one of the following classes: 'basketball_court', 'overpass', 'ground_track_field', 'church', 'chaparral', 'forest', 'parking_lot', 'golf_course', 'baseball_diamond', 'meadow', 'beach','sparse_residential', 'desert', 'terrace', 'palace', 'bridge', 'commercial_area', 'stadium', 'runway', 'lake', 'railway', 'tennis_court', 'ship', 'intersection', 'river', 'freeway', 'airplane', 'industrial_area', 'mountain', 'storage_tank', 'cloud', 'roundabout', 'wetland', 'mobile_home_park', 'island', 'harbor', 'railway_station', 'medium_residential', 'sea_ice', 'thermal_power_station', 'snowberg', 'circular_farmland', 'airport', 'dense_residential', 'rectangular_farmland'. Choose a class from the above classes.

    **Access restrictions:** The dataset is available to downloaded from https://www.kaggle.com/datasets/happyyang/nwpu-data-set

    **Licenses:** No license is provided with the dataset.

    **Ethical considerations:** No personally identifiable information or offensive content is present in the dataset.

26. **GQA** builds up on Visual Genome scene graph structures for reasoning questions. It consists of real-world reasoning, scene understanding, and compositional question answering. Questions are generated using a robust engine which makes sure that the questions are

grounded in the image. Each question is associated with a series of steps that need to be followed to get the answer as well as a scene graph that captures objects, attributes, and relations in the image

**Split:** We use the test split available from https://cs.stanford.edu/people/dorarad/gqa/download.html

**Prompt used:** You are given an image and a question. Answer the question in a single word. Question: *<question>*

**Access restrictions:** The dataset is available to download from https://cs.stanford.edu/people/dorarad/gqa/download.html

**Licenses:** The images in the dataset come with a CC BY 4.0 DEED license https://creativecommons.org/licenses/by/4.0/deed.en

27. **OPENPATH** is a dataset created from Twitter and other public sources. Each image has a natural language description, and the dataset is sourced from tweets across 32 hashtag sub-specialty categories in pathology.

    **Split:** We use the test split for evaluation.

    **Prompt used:** Choose from the below choices, Given image is a hematoxylin and eosin image of: cancer-associated stroma, adipose tissue, debris, lymphocytes, mucus, background, normal colon mucosa, colorectal adenocarcinoma epithelium, smooth muscle

    **Access restrictions:** The dataset is available to download from huggingface datasets https://huggingface.co/datasets/akshayg08/OpenPath

    **Licenses:** The dataset is available under CC BY-NC 4.0 license. https://creativecommons.org/licenses/by-nc/4.0/

    **Ethical considerations:** No personally identifiable information or offensive content is present in the dataset.

28. **IRFL** is an image-text dataset for figurative language. The dataset consists of three broad categories: idioms, similes, and metaphors. Metaphors and similes were collected from online lists whereas idioms were collected from MAGPIE corpus[33]. Since the MAGPIE corpus did not contain definitions for idioms, definitions were crawled from online dictionaries to search for figurative images. Google images were used for searching the images for idioms using these definitions. For similes and metaphors, annotators were used for definitions, and images were searched on the internet. For our evaluation, we use simile categorization. For each data point, one simile and four images are given. We modify this task to evaluate one image at a time, so a pair of an image and similes are passed to the model to see if they match or not.

    **Split:** We use the Simile understanding task for evaluation.

    **Prompt used:** You are given a simile and a picture along with the simile. You have to say if the simile matches the given picture. Answer the following question in a single word with a yes or no. Simile: *<simile>* Answer:

    **Access restrictions:** Dataset is available for download from https://github.com/irfl-dataset/IRFL

    **Licenses:** No license is provided with the dataset.

    **Ethical considerations:** No personally identifiable information or offensive content is present in the dataset.

29. **SCREEN2WORDS** is a mobile UI summarization dataset consisting of images from Rico-SCA[64] dataset. A total of 85 annotators were used to describe the image.

    **Split:** We use the test split from https://github.com/google-research-datasets/screen2words/tree/main

    **Prompt used:** You are given a phone UI screen. Describe the screen in one sentence.

    **Access restrictions:** The dataset is available to download from https://github.com/google-research-datasets/screen2words

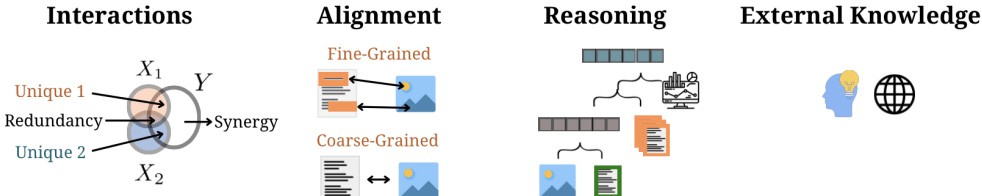

Figure 6: Multimodal skills are the basic building blocks central to solving problems, spanning information integrated across modalities at different granularities, different ways modalities might interact to create new information, reasoning, and external knowledge.

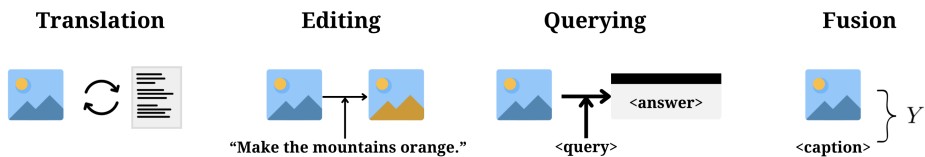

Figure 7: Multimodal information flow studies how the content changes across the two modalities for the task, such as through cross-modal translation, editing, querying, and fusion.

**Licenses:** No license is provided with the dataset.

**Ethical considerations:** No personally identifiable information or offensive content is present in the dataset.

30. **Localized Narratives (COCO subset)** (LNCOCO) was built on images from COCO[71], Flickr30k[119], and ADE20K[127] datasets by annotating these datasets with localized information. We use this dataset for the task of image generation.

    **Split:** We use the COCO subset from the Localized Narratives Dataset [87] containing 8,573 samples. The ground truth images are used from the MSCOCO (17) validation set.

    **Prompt used:** Generate an Image based on the provided caption. Caption:

    **Access restrictions:** The dataset is available to download from https://google.github.io/localized-narratives/

    **Licenses:** The dataset is available under CC BY-NC 4.0 license. https://creativecommons.org/licenses/by-nc/4.0/

    **Ethical considerations:** No personally identifiable information or offensive content is present in the dataset.

## A.2 Dataset Categorization

For categorizing the datasets, we follow a three-stage approach with the majority of the categorizations done using human annotators versed in machine learning, followed by using multimodal large language models to alleviate any annotator disagreement issues, and performing a final check by the authors of this work who are experts in multimodal machine learning.

### A.2.1 Categorization stage 1: Human annotation of dimensions

In the first stage of the annotation process, we sample five data points from each dataset, for a total of 145 data points spread out across 10 sets. Each set was evaluated by two annotators each. Annotators for this task were from the machine learning research community. For each data point, we provide the image, prompt, and the ground truth answer followed by five questions which the annotator has to answer. These questions span across various dimensions which we consider for datasets, which are the following: 1) Does answering this question require you to use external knowledge? [Options: Yes, No] 2) Does answering this question require you to use reasoning? [Options: Less Reasoning, Neutral Reasoning, More Reasoning] 3) Which information flow does the data use? [Options: Querying, Translation, Fusion, Editing] 4) Does the data use fine-grained interactions? [Options: Yes, No] 5) What type of interactions does the data have? [Options: Redundancy, Synergy, Uniqueness]. We calculate inter-annotator agreement for the annotators and present them in Table 6.

Table 6: Inter-Annotator agreement scores for stage 1 annotation.

| Set number | Knowledge | Info. Flow | Interactions | Fine-grained | Reasoning |
|---|---|---|---|---|---|
| 1 | 0.242 | 0.407 | 0.156 | 0.375 | 0.375 |
| 2 | 0.364 | 0.115 | 0.102 | 0.286 | 0.461 |
| 3 | 0.250 | 0.640 | 0.019 | 0.571 | 0.333 |
| 4 | 0.708 | 0.299 | 0.286 | -0.024 | 0.186 |
| 5 | 0.500 | 0.190 | 0.166 | 0.143 | 0.4 |
| 6 | 0.192 | 0.045 | -0.037 | 0.017 | 0 |
| 7 | 0.473 | 0.171 | 0.204 | -0.153 | -0.296 |
| 8 | 0.439 | 0.469 | 0.067 | -0.365 | 0.313 |
| 9 | 0.032 | 0.419 | 0.464 | -0.029 | -0.105 |
| 10 | 0.472 | 0.417 | 0.097 | 0.286 | 0.151 |

Table 7: Categorization after aggregating human annotations.

| Dataset | Knowledge | Reasoning | Info. Flow | Fine-grained | Interactions |
|---|---|---|---|---|---|
| NLVR2 | No | Less | Querying | No | Uniqueness |
| NLVR | No | Less | Querying | Yes | Uniqueness |
| NY CARTOON | Yes | More | Fusion | No | Synergy |
| MM-IMDB | No | Less | Fusion | No | Synergy |
| MEMOTION | Yes | Less | Fusion | No | Redundancy |
| MEMECAP | No | More | Fusion | No | Synergy |
| MAGIC BRUSH | No | Less | Editing | Yes | Synergy |
| IRFL | No | Less | Fusion | No | Synergy |
| HATEFUL MEMES | Yes | Less | Fusion | No | Synergy |
| INATURALIST | Yes | Less | Querying | No | Uniqueness |
| FLICKR30K | No | Less | Translation | No | Uniqueness |
| GQA | No | Less | Querying | Yes | Redundancy |
| ENRICO | Yes | Less | Querying | No | Uniqueness |
| FER-2013 | No | Less | Querying | No | Uniqueness |
| DECIMER | Yes | Less | Translation | Yes | Uniqueness |
| WINOGROUND | No | Less | Querying | Yes | Redundancy |
| VQARAD | Yes | More | Querying | Yes | Uniqueness |
| VQA | No | Less | Querying | Yes | Uniqueness |
| VISUAL GENOME | No | Less | Querying | Yes | Uniqueness |
| VCR | Yes | More | Fusion | Yes | Redundancy |
| UCMERCED LAND USE | Yes | Less | Querying | No | Uniqueness |
| SLAKE | Yes | Less | Querying | Yes | Uniqueness |
| SCREEN2WORDS | No | Less | Translation | No | Uniqueness |
| SCIENCEQA | Yes | Less | Querying | Yes | Synergy |
| RESISC45 | Yes | Less | Querying | No | Uniqueness |
| OPENPATH | Yes | Less | Querying | No | Uniqueness |
| PATHVQA | Yes | Less | Querying | Yes | Uniqueness |
| NOCAPS | No | Less | Translation | Yes | Uniqueness |
| OK-VQA | Yes | Less | Querying | Yes | Uniqueness |
| LNCOCO | Yes | Less | Translation | Yes | Uniqueness |

As per the annotations, we aggregate the annotations for each dataset across each dimension and calculate the maximum occurrence of annotation across all dimensions to categorize the datasets presented in Table 7. We also consider 'Neutral Reasoning' and 'Less Reasoning' to be the same category and label them as 'Less Reasoning' before aggregating over the annotations. However, we see that the inter-annotator scores have low agreement, and some annotations go against the definitions above in the section 2. Hence, we carry out an additional round of the annotation process using GPT-4V, and explain the process below.

### A.2.2 Categorization stage 2: Automatic annotation with human verification

After the first stage was done, we found that most of the annotations were reliable but there were some cases where annotators misunderstood the definitions and tasks which led to low agreement values. For the second stage of the annotation process, we query GPT-4V for categorization of datapoints into dimensions to supplement the human annotations we obtained in the first stage. For each dataset, we consider three samples from each dataset for a total of 87 data points for categorization spread out across six sets. For each data point, we ask the model the same questions as asked to the human annotators above and obtain the categorization across the dimensions. For some questions, the model

Table 8: Categorization after aggregating GPT-4V annotations. Cases where '-' is present are due to the model not providing an answer, citing a lack of information available for evaluating the input. We ignore such cases for categorization.

| Dataset | Knowledge | Reasoning | Info. Flow | Fine-grained | Interactions |
|---|---|---|---|---|---|
| NLVR2 | No | Less | Fusion | Yes | Synergy |
| NLVR | No | More | Querying | Yes | - |
| NY CARTOON | Yes | Less | Fusion | No | Synergy |
| MM-IMDB | No | Less | Fusion | No | Synergy |
| MEMOTION | Yes | Less | Fusion | No | Synergy |
| MEMECAP | Yes | More | Fusion | No | - |
| MAGIC BRUSH | Yes | Less | Editing | No | Synergy |
| IRFL | Yes | Less | Fusion | No | Redundancy |
| HATEFUL MEMES | Yes | More | Fusion | No | Synergy |
| iNATURALIST | Yes | Less | Querying | No | - |
| FLICKR30K | No | Less | Translation | - | - |
| GQA | No | Less | Querying | Yes | Uniqueness |
| ENRICO | No | Less | - | No | - |
| FER-2013 | No | Less | Querying | - | - |
| DECIMER | Yes | More | Translation | No | Uniqueness |
| WINOGROUND | No | Less | Fusion | No | Redundancy |
| VQARAD | Yes | Less | Querying | No | Uniqueness |
| VQA | No | Less | Querying | Yes | Synergy |
| VISUAL GENOME | No | Less | Querying | Yes | - |
| VCR | Yes | Less | Fusion | No | Redundancy |
| UCMERCED LAND USE | Yes | Less | Querying | No | Synergy |
| SLAKE | Yes | More | Querying | Yes | Uniqueness |
| SCREEN2WORDS | No | Less | Fusion | No | - |
| SCIENCEQA | No | Less | Fusion | No | Synergy |
| RESISC45 | No | Less | Querying | - | Uniqueness |
| OPENPATH | Yes | More | Querying | - | - |
| PATHVQA | Yes | Less | Fusion | Yes | - |
| NOCAPS | Yes | Less | Translation | No | Uniqueness |
| OK-VQA | Yes | Less | Querying | Yes | Synergy |
| LNCOCO | Yes | Less | Translation | Yes | Uniqueness |

Table 9: Inter-annotator agreement scores for stage 2 annotations.

| Set number | Knowledge | Info. Flow | Interactions | Fine-grained | Reasoning |
|---|---|---|---|---|---|
| 1 | 0.667 | 0.420 | 0.868 | 0.705 | 0.000 |
| 2 | 0.631 | 0.797 | 0.363 | 1.000 | 0.450 |
| 3 | -0.097 | 1.000 | 0.732 | 0.732 | 0.444 |
| 4 | 0.588 | 0.658 | 0.851 | 0.571 | 0.417 |
| 5 | 0.444 | 1.000 | 0.842 | 0.722 | 0.587 |
| 6 | 0.317 | 1.000 | 0.222 | 0.837 | 0.000 |

refuses to answer the question citing enough information is not provided, so we do not consider the output for categorization. Aggregation is done similarly to stage 1 of the annotation process and the categories are provided in Table 8. For each set, we ask two annotators to label the annotation by GPT-4V as either correct or wrong, depending on the categorization provided by the model. The inter-annotator agreement scores are provided in Table 9. We see improvements over the previous annotation process in some dimensions and datasets, however, cases where annotations do not match the definition persist. Also, GPT-4V does not give output for a few cases due to which aggregation is not possible. Hence, we carry out the third stage of the annotation process to get a more refined categorization.

### A.2.3 Categorization stage 3: Final check by experts

In the third stage of the annotation process, the authors of the project manually go through the annotations from both stages to check for errors and obtain the final categorization of datasets. We present the categorization in Table 10 with the source for each categorization in the table. (1) indicates that the category has been agreed upon both by human annotators and GPT-4V, (2) indicates that GPT-4V better categorizes the dataset for the dimension and hence the annotation from GPT-4V has been chosen, (3) indicates that human annotations better categorize the dataset for the dimension,

Table 10: Final dataset categorization.

| Dataset | Knowledge | Reasoning | Info. Flow | Fine-grained | Interactions |
|---|---|---|---|---|---|
| NLVR2 | No (1) | Less (1) | Querying (3) | No (4) | Redundancy (4) |
| NLVR | No (1) | Less (3) | Querying (1) | Yes (1) | Redundancy (4) |
| NY CARTOON | Yes (1) | More (3) | Fusion (1) | No (1) | Synergy (1) |
| MM-IMDB | No (1) | Less (1) | Fusion (1) | No (1) | Synergy (1) |
| MEMOTION | Yes (1) | More (4) | Fusion (1) | No (1) | Synergy (2) |
| MEMECAP | Yes (2) | More (1) | Fusion (1) | No (1) | Synergy (3) |
| MAGIC BRUSH | No (3) | Less (1) | Editing (1) | Yes (3) | Synergy (1) |
| IRFL | No (3) | More (4) | Fusion (1) | No (1) | Synergy (3) |
| HATEFUL MEMES | Yes (1) | More (2) | Fusion (1) | No (1) | Synergy (1) |
| INATURALIST | Yes (1) | Less (1) | Querying (1) | Yes (4) | Uniqueness (3) |
| FLICKR30K | No (1) | Less (1) | Translation (1) | No (3) | Uniqueness (3) |
| GQA | No (1) | Less (1) | Querying (1) | Yes (1) | Redundancy (3) |
| ENRICO | No (2) | Less (1) | Querying (3) | No (1) | Uniqueness (3) |
| FER-2013 | No (1) | Less (1) | Querying (1) | No (3) | Uniqueness (3) |
| DECIMER | Yes (1) | More (2) | Translation (1) | No (2) | Uniqueness (1) |
| WINOGROUND | No (1) | Less (1) | Querying (3) | Yes (4) | Redundancy (1) |
| VQARAD | Yes (1) | More (4) | Querying (1) | Yes (4) | Redundancy (4) |
| VQA | No (1) | Less (1) | Querying (1) | Yes (1) | Redundancy (4) |
| VISUAL GENOME | No (1) | Less (1) | Querying (1) | Yes (1) | Redundancy (4) |
| VCR | No (4) | Less (2) | Fusion (1) | Yes (3) | Redundancy (1) |
| UCMERCED LAND USE | No (4) | Less (1) | Querying (1) | No (1) | Uniqueness (3) |
| SLAKE | Yes (1) | More (4) | Querying (1) | Yes (4) | Redundancy (4) |
| SCREEN2WORDS | No (1) | Less (1) | Translation (3) | No (1) | Uniqueness (3) |
| SCIENCEQA | Yes (3) | Less (1) | Fusion (4) | No (2) | Synergy (1) |
| RESISC45 | No (2) | Less (1) | Querying (1) | No (3) | Uniqueness (1) |
| OPENPATH | Yes (1) | More (4) | Querying (1) | Yes (4) | Redundancy (4) |
| PATHVQA | Yes (1) | Less (1) | Querying (3) | Yes (4) | Redundancy (4) |
| NOCAPS | No (3) | Less (1) | Translation (1) | No (2) | Uniqueness (1) |
| OK-VQA | Yes (1) | Less (1) | Querying (1) | Yes (1) | Redundancy (4) |
| LNCOCO | Yes (1) | Less (1) | Translation (1) | Yes (1) | Uniqueness (1) |

(4) indicates that authors of this work have categorized the dataset for the dimension. As we can see from Table 10, the majority of categories are agreed upon both by human annotators and GPT-4V, indicating reliability. There are only a few with (4), indicating that authors had to provide the final categorization due to dimensions that were hard to understand by non-experts in multimodal learning and by GPT-4V.

### A.2.4 Details on annotation and participants

The annotations in stages 1 (human annotation) and 2 (automatic inference with human verification) are all university students with some knowledge of machine learning. There were 10 sets of annotations each evaluated by two annotators for a total of 20 annotators. All participation in user studies was voluntary and done for pay at a level consistent with research participation at our university (15 dollars an hour). The annotations in stage 3 (final check) are done by 5 experts in the multimodal machine learning community for a final verification in case of misunderstandings in the first two stages.

### A.3 Modeling categorizations and details

We also evaluate the performance of the models based on various modeling decisions. To achieve this, we categorize the models into various classes based on the following properties:

1. **Interleaved modality training:** In the multi-modal setting, models are broadly trained/fine-tuned either by separately processing individual modalities using modality-specific encoders followed by fusion, or by interleaving the raw modalities first and then processing the interleaved input together.

2. **Instruction Tuning:** Generative multimodal models can be trained/fine-tuned using objectives such as image-text matching, image-grounded text generation [62], etc., to generate relevant outputs. However, recently such generative models are also instruction instruction-tuned in order to generate outputs that resemble human responses. Therefore, we also categorise the models based on whether instruction tuning is employed or not.

3. **Architecture:** For training multi-modal models, parameters can either be initialized using a pre-trained model and then are fine-tuned/kept frozen, or are initialized randomly and trained in an end-to-end fashion. Based on this choice, we categorize models into two classes - fine-tuned and trained from scratch.

4. **Training Data Size:** The amount of data used for training the models, plays an important role in the performance and generalization of the model. Based on the size of the training data (in our work, the number of image-text or image-image samples), we categorize the models into three categories - Small, Medium, and Large.

5. **Number of Parameters:** Model size is an important modeling decision as it affects the performance of the model, cost and efficiency of training, and the inference time. Hence, we also categorize the models based on both the total and trainable number of parameters, and compare the performance across these categories.

6. **Diversity in Training Data:** Training multimodal models on data from different tasks, improves the diversity of the training data and may help the models to perform well on multiple tasks. By categorizing the models based on the diversity of the training data used, we evaluate the effect of using data from diverse tasks.

## A.4 Model Details

For the HEMM benchmark, we currently evaluate the following models. All the models except for Gemini and GPT-4V are open source and we encourage the community to add more models to the benchmark.

1. **BLIP-2** uses pre-trained image encoder and a pre-trained LLM for decoding. A Q-former is used to fuse the input text and the image queries using attention mechanism, and the fused representation is used by the decoder to generate the response. While training, only the parameters of the Q-former are updated using supervised fine-tuning, and the rest of the architecture is kept frozen. In this work we use the `blip2_t5` model with `pretrain_flant5xxl` as the decoder from LAVIS[*]. The chosen model has 108M and 12.1B trainable and total parameters respectively.
   **License:** The model comes with BSD-3 Clause https://github.com/salesforce/LAVIS/blob/main/LICENSE.txt
   **Access restrictions:** The model is available to use from the LAVIS repository https://github.com/salesforce/LAVIS

2. **INSTRUCT-BLIP** is built on top of the BLIP2 architecture, where the model is first pre-trained similar to BLIP2. In the second phase, the Q-former in the architecture is instruction tuned (rest parameters frozen) to create an instruction following Q-former. For evaluation, we use the `blip2_t5_instruct` model with `flant5xl` as the decoder from LAVIS[*]. The model has 188M trainable parameters and 4B parameters in total. The pre-training data for the first phase is similar to BLIP2 and additional 15M samples from diverse datasets and tasks (e.g., VQA, Reasoning, Captioning, etc.) are used for instruction tuning.
   **License:** The model comes with BSD-3 Clause https://github.com/salesforce/LAVIS/blob/main/LICENSE.txt
   **Access restrictions:** The model is available to use from the LAVIS repository https://github.com/salesforce/LAVIS

3. **MINI-GPT-4** also has a similar architecture as BLIP2, and uses the same Vision encoder and Q-former. However, the decoding LLM is based on Vicuna. Further, MiniGPT-4 has an additional single projection layer applied to the output of the Q-former. The architecture is instruction tuned with all the parameters except for the projection layer are kept frozen. We evaluate the `prerained_minigpt4_7b` model from the MiniGPT-4 GitHub repository [*]. The model has 13B parameters and is fine-tuned using 5M image-text samples.
   **License:** The model comes with BSD-3 Clause https://github.com/Vision-CAIR/MiniGPT-4/blob/main/LICENSE.md

---

[*]https://github.com/salesforce/LAVIS/tree/main/projects/blip2

[*]https://github.com/salesforce/LAVIS/tree/main/projects/instructblip

[*]https://github.com/Vision-CAIR/MiniGPT-4?tab=readme-ov-file

**Access restrictions:** The model is available to use from https://github.com/Vision-CAIR/MiniGPT-4/tree/main

4. **OPENFLAMINGO** is an open-source reproduction of the Flamingo [3] models. Unlike models that can only take one input image per sample (e.g., BLIP2, MiniGPT-4), OpenFlamingo can handle multiple images by interleaving images and texts. The architecture comprises of pre-trained Vision and Language encoder/decoder, where the layers of the pre-trained LLM are augmented with the vision encoder outputs which allows for cross-modal attention. All the pre-trained components are kept frozen except for the cross-modal attention component. For evaluation, we use the `OpenFlamingo-3B-vitl-mpt1b` model from the OpenFlamingo Github Repository [*]. The chosen models has 1.4B trainable parameters and a total of 3.2B parameters. It is trained using 180M image-text samples.
   **License:** Work is available under MIT License https://github.com/mlfoundations/open_flamingo/blob/main/LICENSE
   **Access restrictions:** The model is available to use from https://github.com/mlfoundations/open_flamingo

5. **LLAMA-ADAPTER** is based on the architecture of LLaMA Adapter [125] which augments the text tokens with learnable adaptation prompts. In addition to this, LLaMA Adapter V2 uses early fusion to add visual knowledge to the decoding LLM. The architecture uses both early fusion and late fusion, and while fine-tuning, all the pre-trained components are frozen except for the bias layers of the LLM, Visual Projection Layer and the zero-initialized cross attention module. We evaluate the `BIAS-LORA-7B` model which uses LLaMA-7B as the decoder[*]. The model is instruction tuned using 619K samples, and has 14M trainable parameters.
   **License:** Work is available under GNU General public license https://github.com/OpenGVLab/LLaMA-Adapter/blob/main/LICENSE
   **Access restrictions:** Model is available to use from https://github.com/OpenGVLab/LLaMA-Adapter

6. **EMU** is a large multimodal model trained using interleaved video, image and text data, trained in an autoregressive manner to predict the next token in the multimodal sequence. With the ability to produce the next visual token, Emu is also able to generate images and has been evaluated on the Magic Brush dataset in this work. The architecture uses pre-trained encoder and a decoding LLM such as LLaMA. EMU is first pre-trained using interleaved video, image, and text data, and all the parameters are updated during the pre-training. In the second stage, emu is further instruction-tuned. However, in this work we only evaluate the pre-trained version of Emu. We evaluate the Emu-14B model pre-trained using 82M samples.
   **License:** Work is available under Apache 2.0 license https://github.com/baaivision/Emu/blob/main/LICENSE
   **Access restrictions:** The model is available to use from https://github.com/baaivision/Emu

7. **FUYU-8B** is a decoder only architecture where the image patches are linearly projected into the first layer of the transformer architecture. Fuyu's architecture is same as that of Persimmon-8B [*], and we use the details of Persimmon-8B to categorise Fuyu into the model categories. Persimmon-8B has 9.3B parameters and is trained from scratch. In our work we evaluate the pre-trained model as the instruction tuned models aren't available and the pre-training data sources and sizes are unknown. We evaluate the Fuyu-8B model available through HuggingFace [*].
   **License:** Work is available under Creative Commons Attribution Non Commercial 4.0 International license https://spdx.org/licenses/CC-BY-NC-4.0
   **Access restrictions:** Model is available to use from huggingface

---

[*]https://github.com/mlfoundations/open_flamingo

[*]https://github.com/OpenGVLab/LLaMA-Adapter/tree/main/llama_adapter_v2_multimodal7b

[*]https://www.adept.ai/blog/persimmon-8b

[*]https://huggingface.co/adept/fuyu-8b

https://huggingface.co/adept/fuyu-8b

8. **KOSMOS-2** is based on a causal Transformer Language Model, and has the architecture similar to Kosmos1 [40]. It is trained on the next-token prediction task. In addition to the pre-training data used to train Kosmos1, grounded image-text pairs are added to the dataset to train Kosmos2. Overall, Kosmos2 is trained using interleaved image-text data and later instruction-tuned using both multimodal and language-only instructions. We evaluate the `ydshieh/kosmos-2-patch14-224` model from HuggingFace [*] which has a total of 1.6B parameters.
   **License:** Work is available under MIT License https://huggingface.co/datasets/choosealicense/licenses/blob/main/markdown/mit.md
   **Access restrictions:** The model is available to use from https://huggingface.co/microsoft/kosmos-2-patch14-224

9. **mPLUG-OWL** uses a vision foundation model to encode input image and uses a visual abstractor model to summarize the input from the encoder. The abstractor output along with the text queries are then passed to a pre-trained language foundation model that generates the response. The model is first pre-trained using supervised fine-tuning of all the parameters except for the language models. In the second phase, the language models is instruction tuned using multimodal and language instructions, with the other parameters frozen. We evaluate the `https://github.com/X-PLUG/mPLUG-Owl/tree/main/mPLUG-Owl` model obtained from the mPLUG-Owl Github Repository [*]. The chosen models has a total of 7.2B parameters.
   **License:** Work is available under MIT License https://github.com/X-PLUG/mPLUG-Owl/blob/main/LICENSE
   **Access restrictions:** The model is available to use from https://github.com/X-PLUG/mPLUG-Owl

10. **GPT-4V** is a multimodal extension to GPT-4 which has been trained on the next word prediction task using image and text data from the internet and licensed data sources and fine tuned using RLHF[84],[20]. We use 'gpt-4-vision-preview' as a chosen model for our evaluation. As of evaluating the models, 'gpt-4-vision-preview' points to 'gpt-4-1106-vision-preview' in the OpenAI API interface which has been trained up to April 2023 [*].
   **License:** None
   **Access restrictions:** The model is available via OpenAI's API https://platform.openai.com/docs/guides/vision

11. **GEMINI** is a series of multimodal large language models which support interleaved inputs. These models have been trained on multimodal and multilingual data comprising of data from web documents, books, and code, and includes image, audio, and video data. For our evaluation, we use 'gemini-pro-vision' which points to 'gemini-1.0-pro-vision-001' released on February 15, 2024 [*]. We also use safety settings such as 'HARM_CATEGORY_DANGEROUS', 'HARM_CATEGORY_HARASSMENT', 'HARM_CATEGORY_HATE_SPEECH', 'HARM_CATEGORY_SEXUALLY_EXPLICIT', 'HARM_CATEGORY_DANGEROUS_CONTENT' and set the threshold to 'BLOCK_NONE' provided by the API [*].
   **License:** None
   **Access:** Available via API https://ai.google.dev/gemini-api/docs/models/gemini

---

[*] https://huggingface.co/microsoft/kosmos-2-patch14-224

[*] https://github.com/X-PLUG/mPLUG-Owl/tree/main/mPLUG-Owl

[*] https://platform.openai.com/docs/models/gpt-4-turbo-and-gpt-4

[*] https://cloud.google.com/vertex-ai/generative-ai/docs/learn/model-versioning

[*] https://ai.google.dev/gemini-api/docs/safety-settings

Table 11: Evaluation metrics supported in HEMM

| Metric | Task | Modalities |
|---|---|---|
| BLEU | Text Generation | Text |
| ROUGE | Text Generation | Text |
| BertScore | Text Generation | Text |
| BARTScore | Text Generation | Text |
| RefCLIPScore | Text Generation | Image, Text |
| CLIP-I | Image Generation | Image |
| MSE | Image Generation | Image |

# B  Experimental Details

## B.1  Evaluation metrics

We present our results on BARTScore [122] as models under our evaluation generate noisy free from text, however, we also support other text generation metrics under our evaluation suite listed in the Table 11 below.

## B.2  Evaluation protocol

HEMM supports image generation tasks, models and metrics. However, currently there are only 2 image generation tasks (LNCOCO and MAGIC BRUSH) and 1 model (EMU) that supports image generation. Hence, we perform all our evaluation on the remaining 28 text generation tasks and report the results on the image generation tasks in Appendix C.

Note: Since HEMM contains models that are unable to process multiple images in the same input, we modify the WINOGROUND and IRFL tasks (as per A.1) in order to have a single image-text pair as input for each sample.

For each dataset, we use the same prompts across all models as shown in Section C, for standardization, however, there can be a scenario where these models perform better with other prompts or scenarios and may perform poorly under our scenarios or prompts in our evaluation.

For each dataset, the computed metrics for the models are normalized on a scale of 0 to 1, 0 corresponds to the model achieving the lowest score on that dataset, and 1 corresponds to the performance achieve by exactly generating the ground truth. For BERTScore [126], ROUGE [70], and RefCLIPScore [36] the maximum value is set to 1. BARTScore [122] uses the log of probabilities. Following [16], we calculate the maximum value for each dataset separately as BARTScore(r, r) where r is the ground truth sentence.

Since details regarding training type for GEMINI and GPT-4V, and modality processing for GPT-4V are not revealed, we do not use the scores from these models while evaluating the performance for the training type and modality processing dimensions. Further, for HATEFUL MEMES, OPENPATH, and MEMOTION datasets, GPT-4V did not respond and generated *can't provide assistance* and *"indeterminate"* for many samples. Hence, we exclude the results of GPT-4V on these datasets during evaluation.

## B.3  Significance tests

While comparing performance across categories in each dimension, we perform paired t-tests to determine the significance of the results. For datasets, specifically, for each category, we calculate the average performance of each of the 11 models on all the datasets in a category ($c_i$) to create a vector $v_i \in \mathbb{R}^{11}$. Next, we performed pairwise t-tests between these vectors to determine the significance of the results. The p-values obtained through the t-tests are presented in Table 13. We find that the difference between the performance on different categories is statistically significant (p-value $< 0.05$) for real-world use cases, multimodal interaction, external knowledge, and information flow dimensions, which explains that these are particularly difficult dimensions for today's multimodal model.

We also conducted t-tests for various categories in each of the modeling dimensions. For all models in a category ($c_i$), we use their average performance on each of the 28 datasets to construct a vector $w_i \in \mathbb{R}^{28}$. We then perform pair-wise t-tests across all the categories for all dimensions. As mentioned

Table 12: Hyperparameters used for running inference for various models. Temperature for GPT-4V and Beam Size for GPT-4V and GEMINI are unknown. We also report the average inference time in seconds for an image-text input. For each models, we take the average of inference times across all the datasets.

| Model | Temperature | Beam Size | Max New Tokens | Inference Time |
|---|---|---|---|---|
| BLIP-2 | 1.0 | 5 | 30 | 0.64 |
| INSTRUCT-BLIP | 1.0 | 5 | 256 | 0.58 |
| MINI-GPT-4 | 1.0 | 3 | 100 | 11.8 |
| FUYU-8B | 1.0 | 1 | 100 | 1.92 |
| EMU | 0.9 | 5 | 100 | 1.43 |
| OPENFLAMINGO | 1.0 | 3 | 50 | 2.35 |
| KOSMOS-2 | 1.0 | 1 | 500 | 0.31 |
| MPLUG-OWL | 1.0 | 1 | 100 | 0.87 |
| LLAMA-ADAPTER | 0.0 | 1 | 100 | 1.30 |
| GPT-4V | - | - | 300 | 2.67 |
| GEMINI | 0.4 | - | 2048 | 4.62 |

in Section B.2, we do not use the scores of GPT-4V and GEMINI for the dimensions where their training/modeling decisions aren't revealed. We find that for all the dimensions, the best-performing category achieves significantly better scores with p-values $< 0.05$ (Table 14).

## B.4 Model hyperparameters and inference time

In Table 12, we list the values of important text-generation hyperparameters used to evaluate different models. For each model, we also report the inference time for a single image-text pair averaged across all the datasets.

## B.5 Human evaluation

We perform human preference-based pair-wise comparison (battles) of model responses across 1000 datapoints and use the following metrics to rank the models.

**Average win rate:** Similar to Chiang et al. [19], for each pair of models, considering only the battles between them, we determine the win rate $w_{ab} = \frac{N_a}{N_a + N_b}$, where $N_a$ and $N_b$ are the number of battles won by $model_a$ and $model_b$ respectively. We then take the average of the win rates across all the models to calculate the average win rate for each model i.e., $awr_a = \frac{1}{M} \sum_{b=1}^{M} w_{ab}$.

The top 4 models based on the average win rate are GEMINI (0.73), GPT-4V (0.68), INSTRUCT-BLIP (0.60) and BLIP-2 (0.52).

**Elo Rating:** Using the initial rating of each model as 1000, we sequentially process the battles and update the rating of the models as per the below equations. $R_a$ and $R_b$ denote the current ratings of $model_a$ and $model_b$ in the battle. $S_a = 1$ if $model_a$ wins the battle and 0 if it loses. $S_b = 1 - S_a$ and in case of ties, $S_a = S_b = 0.5$. For more stable Elo ratings, we use K = 4.

$$E_a = \frac{1}{1 + 10^{(R_b - R_a)/400}}; \quad E_b = \frac{1}{1 + 10^{(R_a - R_b)/400}}$$
$$\hat{R}_a = R_a + K * (S_a - E_a); \quad \hat{R}_b = R_b + K * (S_b - E_b)$$

The above update rule is sensitive to battle orders. In order to get more stable and less biased Elo ratings, we run the above computation 1000 times by shuffling the battle order each time, and report the median Elo rating over the 1000 runs for each model.

The 1000 battles were split across 5 authors randomly (200 battles each) for annotation. Using a web interface, the model outputs were presented to the annotators. For each sample, the annotators were instructed to select the output that better answers the query. For cases where both outputs were equally good/bad, or performing the task required domain knowledge (e.g., healthcare datasets), the annotators were instructed to choose the Tie option. For each battle, the models were anonymized for fair comparison.

Table 13: Standard deviation and p-values (from paired t-tests) across categories for each dataset dimension. On average, models achieve significantly higher scores on Multimedia and Affect as compared to other use cases. The p-values for Reasoning and Granularity dimensions are higher than 0.05, indicating that there is no category significantly more challenging than the rest.

| Dimension | Category | Perf (↑) | P-value |
|---|---|---|---|
| Real-world use case | **Multimedia** | **31.30 ± 0.14** | vs Affect: 0.1100
vs Health: 0.0006
vs Science: 0.0000
vs HCI: 0.0002 |
| | Affect | 30.35 ± 0.15 | vs Health: 0.0044
vs Science: 0.0018
vs HCI: 0.0011 |
| | Health | 20.24 ± 0.09 | vs Science: 0.8806
vs HCI: 0.0961 |
| | Science | 19.83 ± 0.13 | vs HCI: 0.2093 |
| | HCI | 15.70 ± 0.08 | |
| Multimodal interaction | Redundancy | 29.04 ± 0.14 | vs Uniqueness: 0.0008
vs Synergy: 0.0522 |
| | Uniqueness | 19.60 ± 0.10 | vs Synergy: 0.0000 |
| | **Synergy** | **33.73 ± 0.15** | |
| Reasoning | More Reasoning | 27.50 ± 0.11 | vs Less Reasoning: 0.6415 |
| | Less Reasoning | 26.84 ± 0.13 | |
| Granularity | Fine-grained | 26.52 ± 0.12 | vs Coarse-grained: 0.5887 |
| | Coarse-grained | 27.52 ± 0.13 | |
| Knowledge | External Knowledge | 23.51 ± 0.10 | vs None: 0.0023 |
| | **None** | **29.62 ± 0.14** | |
| Information flow | Querying | 25.88 ± 0.13 | vs Translation: 0.0479
vs Fusion: 0.0018 |
| | Translation | 18.97 ± 0.07 | vs Fusion: 0.0004 |
| | **Fusion** | **33.77 ± 0.15** | |

# C   All Results

Due to query limits for GPT-4V and GEMINI, we evaluated the two models only on 100 samples per dataset, and for a fair comparison, we performed our analysis using the outputs of all the models on those 100 samples. In this section, we present the results and analysis on the whole evaluation set using the outputs of all the models except GPT-4V and GEMINI. Further, since our analysis was based on text-generation tasks, we present here the results on the image-generation tasks - MAGIC BRUSH and LNCOCO. Specifically, we evaluated EMU (only model in HEMM that can generate images) on both tasks. We find the MSE and the CLIP-I score between the generated and the ground truth image for MAGIC BRUSH to be 0.17 and 0.54. For the LNCOCO dataset, the MSE and CLIP-I score are 0.18 and 0.50.

Note: due to high inference time of some models (e.g., MINI-GPT-4, EMU, OPENFLAMINGO), missing image URLs in the NLVR2 dataset, and compute restrictions for larger evaluation sets like MM-IMDB, VISUAL GENOME, and INATURALIST, we use the results from the same 100 samples used for evaluation in Section 4.

## C.1   Dataset and model comparisons

**Dataset comparisons:** On average, the models achieve the highest scores on IRFL (0.53), WINOGROUND (0.42), and NLVR (0.40) datasets. Healthcare, Science, and HCI datasets are the most challenging use cases for the models with the average scores being the lowest for DECIMER (0.05),

Table 14: Standard deviation and p-values for categories in various modeling dimensions. Models in the best-performing category in each dimension, receive significantly higher scores than the other categories.

| Dimension | Category | Perf (↑) | P-value |
|---|---|---|---|
| Modality Processing | Interleaved | 22.94 ± 0.10 | vs Separate: 0.0011 |
| | **Separate** | **28.58 ± 0.15** | |
| Model Size | Small | 23.34 ± 0.14 | vs Medium: 0.7370 vs Large: 0.0004 |
| | Medium | 23.87 ± 0.12 | vs Large: 0.0004 |
| | **Large** | **42.33 ± 0.07** | |
| Training Type | **Modular** | **24.92 ± 0.12** | vs End-to-End: 0.0427 |
| | End-to-End | 21.26 ± 0.13 | |
| Size of Training Data | Small | 16.80 ± 0.10 | vs Medium: 0.0000 vs Large: 0.0000 |
| | Medium | 30.10 ± 0.15 | vs Large: 0.5024 |
| | **Large** | **31.77 ± 0.16** | |
| Diversity of Training Data | Non-diverse | 21.71 ± 0.12 | vs Diverse: 0.0000 |
| | **Diverse** | **30.15 ± 0.14** | |
| Instruction Tuning | No | 22.49 ± 0.11 | vs Yes: 0.0004 |
| | **Yes** | **29.71 ± 0.15** | |

PATHVQA (0.06), INATURALIST (0.06), and ENRICO (0.08). Meme datasets are also challenging for the models. A low average score (0.12) on MEMECAP shows that the models struggle to understand the visual metaphors and generate suitable captions for the memes.

**Model comparisons:** Overall, INSTRUCT-BLIP and BLIP-2 achieve the highest average scores of 0.38 and 0.37, followed by FUYU-8B (0.29). OPENFLAMINGO and EMU rank lowest on many datasets (receiving a 0 score as per our normalization) and achieve the lowest average scores of 0.05 and 0.11.

### C.2    Dataset trends

In Table 15, we summarize the average performance of models on various categories in each data dimension. We now closely compare the performance between different categories of individual dimensions.

**Multimodal Skills 1: Interactions** The average scores on datasets having redundant, unique and synergistic interactions are 0.25, 0.14, and 0.28. The p-values obtained using paired t-test for Redundancy vs Uniqueness, Uniqueness vs Synergy, and Redundancy vs Synergy are 0.01, 0.0008, and 0.22, indicating that average scores on datasets with unique interactions is significantly lower as compared to datasets with Redundant and Synergistic interactions. Reasons for lower uniqueness scores can be attributed to the presence of highly challenging datasets such as DECIMER, INATURALIST, ENRICO.

**Multimodal Skills 2: Granularity** The average scores of the models on datasets with fine-grained (0.23) and coarse-grained alignment (0.22) are not significantly different, indicating that both categories are challenging for the models, with the former containing tasks like GQA, WINOGROUND and NLVR and the latter having tasks such as FLICKR30K, HATEFUL MEMES, and SCIENCEQA.

**Multimodal Skills 3: Reasoning** The average scores achieved by models on tasks requiring less or more reasoning are 0.22 and 0.23 respectively, and we find that the difference is not statistically significant. This indicates that both categories are challenging for the models with the less reasoning category comprising of datasets like ENRICO and INATURALIST posing challenges related to visual perception and external knowledge. On the other hand, tasks within the more reasoning category such as VCR and MEMECAP test for compositional and commonsense reasoning.

Table 15: Comparisons on different dataset categories. 30 multimodal datasets are split into various groups based on their real-world use case, type of multimodal interaction, presence of reasoning and external knowledge, granularity of alignment, and types of information flow. Performance is measured via the mean BARTscore across 9 multimodal models.

| Category | Group | Perf (↑) |
|---|---|---|
| Real-world use case | Multimedia | **29.27 ± 0.14** |
| | Affect | 22.63 ± 0.11 |
| | Health | 15.51 ± 0.08 |
| | Science | 14.23 ± 0.08 |
| | HCI | 12.49 ± 0.07 |
| Multimodal interaction | Redundancy | 24.86 ± 0.13 |
| | Uniqueness | 13.87 ± 0.06 |
| | Synergy | **28.48 ± 0.13** |
| Reasoning | More | 23.19 ± 0.11 |
| | Less | 21.78 ± 0.09 |
| Granularity | Fine-grained | 22.97 ± 0.11 |
| | Coarse-grained | 21.68 ± 0.10 |
| Knowledge | External | 19.60 ± 0.09 |
| | None | **24.21 ± 0.11** |
| Information flow | Querying | 20.15 ± 0.10 |
| | Translation | 16.72 ± 0.07 |
| | Fusion | **29.16 ± 0.14** |

Table 16: Comparisons on different modeling decisions. We group models based on the modeling and training decisions, including how they process modalities, their parameter counts, model architecture, training data size and diversity, and the presence of instruction tuning. Performance is measured via the mean BARTscore across all 30 tested multimodal datasets.

| Category | Group | Perf (↑) |
|---|---|---|
| Modeling decisions | | |
| Modality processing | Interleaved | 16.92 ± 0.09 |
| | Separate | **26.48 ± 0.15** |
| Model size | Small | 21.51 ± 0.13 |
| | Medium | 22.59 ± 0.12 |
| Training decisions | | |
| Training type | Modular | 23.18 ± 0.13 |
| | End-to-end | 20.93 ± 0.13 |
| Size of training data | Small | 16.08 ± 0.11 |
| | Medium | **27.60 ± 0.15** |
| | Large | 20.72 ± 0.15 |
| Diversity of training data | Non-diverse | 19.92 ± 0.12 |
| | Diverse | **24.09 ± 0.13** |
| Instruction tuning | No | 21.00 ± 0.12 |
| | Yes | **23.22 ± 0.14** |

**Multimodal Skills 4: External Knowledge** Average performance of models on tasks requiring external knowledge (0.20) is significantly lower than tasks not requiring knowledge (0.24). For example, on average, models perform better on NLVR, FER-2013 and WINOGROUND that do not require external knowledge as compared to tasks like iNATURALIST and SLAKE which require external knowledge to identify appropriate species or organs in the image.

**Multimodal Skills 5: Information flow** Models achieve significantly lower average score on translation datasets (0.17) as compared to querying (0.20) and fusion (0.29) datasets. Lower scores on translation dataset is due to the presence of highly challenging datasets such as DECIMER which requires domain knowledge of molecules to generate the correct textual sequence.

### C.3 Modeling trends

**Model scale:** Since we do not consider GPT-4V and GEMINI for analysis in this section, there are no models in the large category. Amongst small and medium models, we find no significant difference (p-value = 0.45) between the average performance of models from the two categories with small and medium models receiving 0.21 and 0.23 average scores respectively.

**Pretraining data scale:** On average, models with medium pretraining data achieve the highest score (0.28) as compared to the models pretrained with small (0.16) or large (0.21) scale data. Although the average score of models trained with large pretraining data is lower as compared to models trained with medium pretraining data, we find that the former models perform better on tasks such as IRFL, WINOGROUND, MEMECAP, and DECIMER which require complex reasoning and external knowledge.

**Diversity of pretraining data:** Models trained with diverse pretraining data (0.24) perform better than models trained only on image-captioning datasets (0.20). The p-value for the paired t-test is 0.01 indicating that the difference is significant. On average, we find that models pretrained with diverse data achieve better scores on knowledge-intensive tasks such as iNATURALIST and OK-VQA with improvements in average scores up to 0.21.

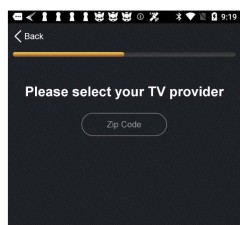 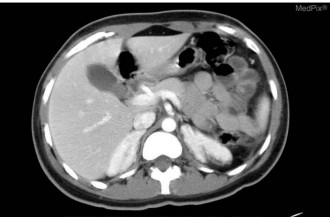 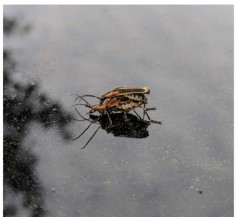 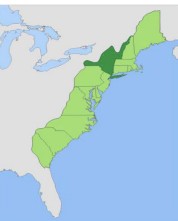

**Prompt:** Given a screenshot of the user interface of a mobile application. Choose the most appropriate design topic from the following comma separated choices:......Answer:
**GPT-4V:** Form
**Gemini:** TV provider selection
**InstructBLIP:** A screen that says please select your tv provider
**Ground Truth:** Search

**Prompt:** Given a radiology image and a question. Answer the question in a single word. Question: Where are the kidney? Answer:
**GPT-4V:** Flanks
**Gemini:** Posterior
**InstructBLIP:** Left
**Ground Truth:** Not Seen Here

**Question:** The scientific species name of the species present in the image is:
**GPT-4V:** Blattella germanica
**Gemini:** Lytta magister
**InstructBLIP:** Tephritidae
**Ground Truth:** Chauliognathus marginatus

**Question:** What is the name of the colony shown?
**GPT-4V:** New Hampshire
**Gemini:** Vermont
**InstructBLIP:** New Hampshire
**Ground Truth:** New York

(a)            (b)            (c)            (d)

Figure 8: Model outputs on samples from ENRICO, VQARAD, INATURALIST, and SCIENCEQA. In (a), all the models struggle to reason about the use of the *zip code* field in the UI, which will be used to *search* the TV provider. Example (b) underscores the complexity faced by models in interpreting medical images, particularly evident in their inability to recognize the absence of a kidney in the radiology image. As shown in (c), the highly fine-grained INATURALIST dataset is very challenging and none of the models can determine the species of the insect. In (d), all models provide incorrect responses when tasked with identifying the colony's name, illustrating the challenges posed by tasks requiring external knowledge.

**Instruction tuning vs supervised fine-tuning:** Instruction-tuned models achieve a higher average score (0.23) as compared to models with only supervised fine-tuning (0.21). We observe the highest improvements in translation tasks such as DECIMER, FLICKR30K, and SCREEN2WORDS. We also observe that instruction-tuned models receive a higher average score as compared to supervised fine-tuned models (improvement of 0.12).

**Modality processing:** Models that process the modalities separately perform significantly better than the models that operate on interleaved inputs. The average scores for the former and latter models are 0.17 and 0.26 respectively (p-value $\approx$ 0). We find high improvements of 0.26, 0.24, 0.22, and 0.2 in the average scores for the datasets SCIENCEQA, NY CARTOON, MM-IMDB, and UCMERCED LAND USE.

**Training type:** We do not find a significant difference between the models that are fine-tuned in a single phase end-to-end manner (0.21) as compared to the models where only specific modules are fine-tuned in a single phase (0.23).

### C.4 Summary of takeaway messages

Finally, we summarize the main findings regarding the performance and evaluation of multimodal foundation models that can be important directions for future work:

1. **Challenging datasets**: Health, HCI, and Science are all relatively difficult use cases for today's multimodal foundation models, which are statistically significantly harder than Multimedia and Affective Computing use cases. In particular, images of scientific diagrams, satellite images, medical images, memes, and rich social interactions pose challenges. It is therefore important to evaluate multimodal models on a diverse range of input modalities and output tasks to get a better measure of generalization performance.
2. **Multimodal interactions**: Models perform better on redundant interactions but struggle when visual information is not directly referenced by text (i.e., uniqueness or synergy). Future benchmarks should contain richer multimodal interactions beyond redundancy, such as in analyzing sarcasm, humor, memes, science, environment, and education. These can serve as better test beds for multimodal models and enable their applications towards real-world multimodal interactions.
3. **Reasoning, fine-grained, and knowledge**: We need better datasets that test for complex reasoning and fine-grained alignment - current ones do not pose enough challenges to today's models, with no significant performance differences with or without reasoning and fine-grained alignment.

We do find that tasks requiring external knowledge are significantly harder than no knowledge; bridging this gap can be a promising direction for multimodal research.

4. **Model and data size**: Perhaps unsurprisingly, larger scales of data and models improve the average score across the board, with significant improvements of up to 75% as compared to medium-sized models. Training on diverse data sources also improves over models that only pretrain on images and captions. The tasks that show the most improvement are INATURALIST and MEMECAP which are knowledge-intensive and require complex reasoning.

5. **Model training**: Instruction-tuned models performed better than those with only supervised fine-tuning. Cross-modal translation (image-to-text) tasks show the most improvements (e.g., DECIMER, MEMECAP, and SCREEN2WORDS). However, some instruction-tuned models still struggle to follow the instructions (e.g., generating a caption when asked to classify an image, or generating long responses when asked to answer in a few words). Instruction tuning using larger datasets with diverse instructions can help alleviate this problem.

