# OpenReview forum: "HEMM: Holistic Evaluation of Multimodal Foundation Models"
_NeurIPS.cc/2024/Datasets_and_Benchmarks_Track — NeurIPS 2024 Track Datasets and Benchmarks Poster_

### Official Review · Reviewer_PdCU · 2024-06-29

**Rating:** 6
**Confidence:** 4
**Correctness:** Yes
**Clarity:** Yes

**Review:**

Questions:

1. The URL on Line53 is invalid, so I can not find the dataset.

2. Why the three dimensions basic skills, information flow, and real-world use cases can better evaluate the multimodal foundation models.

Cons:

1. The main work of this manuscript is to reorganize and classify existing data sets without introducing new data sets, which limits its contribution.

2. Lack of many representative multimodal models, like LLaVA, Qwen-VL, Claude, etc.

3. The colors in Figure 3 are similar, making it difficult for readers to clearly identify the corresponding models.

4. There is a lack of comparison with other related works. After all, there are already a lot of work related to multimodal model evaluation.

**Strengths:**

1. Systematically evaluate the capabilities of multimodal foundation models is important and valuable.

2. The proposed assessment dimensions including basic skills, information flow, and real-world use cases make sense.

**Additional Feedback:**

None

**Documentation:**

I can not find the datasets.

**Limitations:**

Yes

**Opportunities For Improvement:**

In addition to the questions and cons above, some suggestions can be considered for a better presentation:

1. The leaderboard table containing detailed values ​​should be displayed instead of Figures 3 and 5 which only contain rough information.

2. It is recommended that the data in Table 1 be shown in the appendix accordingly with legends to let readers know more intuitively what the tasks are.

**Relation To Prior Work:**

Insufficient

**Summary And Contributions:**

This manuscript presents Holistic Evaluation of Multimodal Models (HEMM) to systematically evaluate the capabilities of multimodal foundation models across a set of 3 dimensions: basic skills, information flow, and real-world use cases. 30 tasks are included in HEMM. 11 representative models are evaluated on HEMM.

---

> ### Author Rebuttal · Authors · 2024-08-17
>
> Dear reviewer:
> Thanks for the constructive and insightful comments. We have addressed your comments below. Please feel free to follow up if you have further questions!
>
> **[url]** The link to our benchmark can be found at
> https://anonymous.4open.science/r/hemm-save-06D5/. It includes all data loading, model training, and evaluation code. It also includes instructions for using HEMM, contributing datasets and models, and submitting to the leaderboard. HEMM will continue to be maintained and extended for the research community. We had previously included the HEMM code in the zipped supplementary material.
>
> **[3 dimensions]** We believe that the 3 dimensions of basic skills, information flow, and real-world use cases comprehensively taxonomize different parts of the multimodal learning pipeline. At the data level, basic skills such as multimodal interactions and reasoning are inherent properties of different multimodal data distributions. At the feature level, information flow measures how latent representations are designed to fuse, translate, or generate information. Finally, at the prediction level, real-world use cases describe the unique tasks and challenges involved when making predictions. This framework enables us to systematically navigate the space of multimodal datasets and results in HEMM being the most comprehensive benchmark for multimodal foundation models, with extensive coverage across data, feature, and prediction-level challenges while minimizing the overlap between overly similar datasets.
>
> **[new datasets]** The main contribution of HEMM is the taxonomy of basic skills, information flow, and real-world use cases that systematically cover the space of multimodal datasets. Furthermore, our standardized data loading, model training, and evaluation pipelines all significantly reduce the barrier of entry to training and evaluating multimodal models, enabling reproducible and reliable progress. To the best of our knowledge, there has not been a similarly motivated and realized benchmark for multimodal foundation models. If multimodal researchers want to benchmark their new models on medical or satellite imaging datasets, it would incur significant effort, whereas HEMM makes that extremely simple through a few lines of code.
>
>
> **[new models]** We have added LLaVA-1.6, Qwen-VL-chat, InternVL, and Gemini-1.5-flash that have come out recently. We provided some preliminary results and observations with the latest added models in **Figure 1** in the [PDF](https://openreview.net/attachment?id=ffOsqhBtqw&name=pdf) that we provided.
>
> For the rebuttal, we present initial results on 5 representative datasets that cover the space of multimodal use cases (Memecap and NewyorkerCartoon for affective computing and multimedia, Decimer for science, PathVQA for healthcare, and Screen2Words for HCI). We will include the full results with LLaVA-1.6, Qwen-VL-chat, InternVL, and Gemini-1.5-flash in our revised paper. We have also included updated plots and trends of performance in the rebuttal pdf and will add these summary statistics to the main paper.
>
> The results show that Gemini-1.5-flash slightly outperforms other recent models like InternVL, LLaVA-1.6, and Qwen-VL-chat on tasks such as Memecap, Screen2Words, Decimer, and NewyorkerCartoon. However, it performs slightly worse than these models on the PathVQA dataset, where InternVL is strong. One standout finding is that Gemini-1.5-flash achieves significantly higher performance on Decimer, a dataset focused on chemical domain question answering. It far surpasses other models, with scores like InternVL (0.157), LLaVA-1.6 (0.062), and Qwen-VL-chat (0.09). This indicates that Gemini-1.5-flash is particularly well-trained in scientific and specialized data.
>
> Additionally, we observe that newer models generally perform better than older models like InstructBLIP, Mini-GPT-4, and Gemini-1.0 across most tasks, highlighting the overall advancement in multimodal abilities in recent models. Based on new models, HEMM shows progress in Vision-Language learning and also helps identify tasks that are more challenging for state-of-the-art models.
>
>
>
> **[Figure 3]** We have replaced Figure 3 with more friendly colors and included it in the rebuttal pdf. Please refer to the rebuttal pdf in the overall rebuttal part for details of Figure 3. Since Figure 3 includes too many models and can be unclear to read, we also split models based on their model size and made it into 3 subgroups to plot.
>
> **[related works]** We have expanded upon our related works section and included comparisons with some recent multimodal benchmarks for foundation models. MMT-Bench [1] and MMMU [2], CV-Bench[3] are recent multimodal benchmarks that evaluate models on visual recognition, real-world use cases (e.g., navigation), college-level subject knowledge, etc. However, these benchmarks do not evaluate the models on various multimodal skills (e.g., alignment, knowledge) that are essential to perform the tasks. Datasets in HEMM are curated to test these multimodal skills including basic skills, information flow, and real-world use cases.
> .
>
> **[leaderboard]** We are developing a leaderboard together with huggingface team to provide a convenient multimodal evaluation. This leaderboard will make it easier for users to evaluate their models by submitting prediction results and receiving scores automatically. Our goal is to provide a streamlined and accessible platform where users can compare their models and track their performance against others.
>
> **[Table 1]** We have replaced Table 1 with a summary of the tasks and moved the full detailed table to the appendix.
>
> continue with citation

---

> > ### Author Rebuttal · Authors · 2024-08-17
> >
> > [1] Ying, Kaining, et al. "Mmt-bench: A comprehensive multimodal benchmark for evaluating large vision-language models towards multitask agi." arXiv preprint arXiv:2404.16006 (2024).
> >
> > [2] Yue, Xiang, et al. "Mmmu: A massive multi-discipline multimodal understanding and reasoning benchmark for expert agi." Proceedings of the IEEE/CVF Conference on Computer Vision and Pattern Recognition. 2024.
> >
> > [3] Tong, Shengbang, et al. "Cambrian-1: A fully open, vision-centric exploration of multimodal llms." arXiv preprint arXiv:2406.16860 (2024).

---

> > ### Comment · Reviewer_PdCU · 2024-08-21
> > **I still can not open the link**
> >
> > Thanks the rebuttal from the authors, the link above is invalid for me, Is it a problem with my network?

---

> > > ### Author Response · Authors · 2024-08-26
> > > **follow-up to rebuttal**
> > >
> > > Hi reviewer, we were wondering if you had any additional concerns after our rebuttal. Please let us know and we will be happy to engage further.

---

> > ### Author Response · Authors · 2024-08-21
> >
> > Hi reviewer, we have thoroughly reviewed the links and wish to clarify the following:
> >
> > 1. Regarding the link to the code repository: https://anonymous.4open.science/r/hemm-save-06D5/: We tested this link with multiple users across different environments and platforms, and it consistently works for us. If there are any specific issues encountered, we would appreciate further details to troubleshoot the problem effectively.
> >
> > 2. Regarding the PDF link: https://openreview.net/attachment?id=ffOsqhBtqw&name=pdf: We also tested this link and found that it works reliably, as long as it is accessed directly through a standard browser once you logged in with your openreview account. If there are any particular technical or access challenges, please let us know so that we can investigate and address them.
> >
> >
> > We are committed to ensuring the accessibility and transparency of our work. If you continue to encounter issues, we are happy to provide alternative solutions, such as a different link or direct sharing of the files.

---

> ### Comment · Reviewer_PdCU · 2024-08-31
> **Incline to accept, adjust the score from 4 to 6**
>
> Dear Authors:
>
> Maybe it's because of my internet connection that I still can't open the link above, but I find your code at https://github.com/pliang279/HEMM, so my main concern is resolved. Also, I acknowledge your clarification of the contribution in the above rebuttal. However, the presentation problem should be fixed in the revised version, including the similar color in Figure 3, lacking of detailed leaderboard value, blurred examples in Figure 2. Based on the above considerations, I am inclined to accept and adjust the score from 4 to 6.
>
> I can not edit my review now, but I will still give my comments during the discussion stage.
>
> Good luck!

---

> ### Comment · Reviewer_PdCU · 2024-09-02
> **comment**
>
> I can edit my review now, I have updated my score.

---

### Official Review · Reviewer_VBn8 · 2024-07-21
**Interesting and challenging benchmark for evaluating multimodal foundation models**

**Rating:** 7
**Confidence:** 4
**Correctness:** Yes. The evaluation methods and exper…
**Clarity:** Yes. The paper is very well written a…

**Review:**

This paper presents an interesting taxonomy and a thorough benchmark to evaluate multimodal foundation models. The different performances of models on different tasks and dimension shows the significance of this benchmark.

**Strengths:**

1. The paper is organized and presented very well and is easy to follow.
2. The systematic evaluation proposed in this paper helps with measuring the progress in multimodal learning research. In turn, this can help with the development of novel multimodal foundation models.
3. The results show that state-of-the-art foundation models struggle with some of the tasks which suggests this benchmark is challenging and should be interesting for the community.
4. Table 4 which shows performance based on different modeling decisions is very insightful.

**Additional Feedback:**

N/A

**Documentation:**

On line 53, the authors mention "HEMM is publicly available at anon," but there is no link. Therefore, I am not sure whether the code and data are publicly available or not.

Regarding the maintenance plan, on lines 342 and 343 the authors mention "HEMM is publicly available, will be regularly updated, and encourages community involvement in its expansion."

**Ethics:**

No. I don't see any ethical concerns in this work. However, the authors use published models and datasets that might have their innate risks and ethical considerations such as problems with LLMs.

**Limitations:**

Yes, the authors talk about the limitations and potential societal impact of this work.

**Opportunities For Improvement:**

Not a limitation, but it would have been interesting to see how much one-shot or few-shot prompting would change the performance on HEMM.
Also, having a live website of the benchmark would be very useful. Maybe the authors already have it, but the link they provided in the paper did not work.

The main argument against this paper would be that the benchmark consists of already existing datasets and the authors do not create anything new. However, I think the novelty lies in the taxonomy and a holistic evaluation of multimodal foundation models instead of reporting performance on a single task/dataset.

**Relation To Prior Work:**

The authors position their work very well with respect to the previous benchmarks and datasets in the area of multimodal learning.

**Summary And Contributions:**

The authors present the Holistic Evaluation of Multimodal Model (HEMM) a taxonomy and systematic evaluation framework of multimodal foundation models which makes it easier to compare models and datasets.
HEMM considers three dimensions for evaluation; basic multimodal skills, multimodal information flow, and real-world use cases. These three dimensions cover 30 tasks (30 existing datasets such as VQA and VISUAL GENOME).

Basic multimodal skills include multimodal interactions, granularity of multimodal alignment, and reasoning and external knowledge.
Multimodal information flow includes cross-modal translation, cross-modal editing, cross-modal querying, and multimodal fusion.
Real-world use cases include multimedia, affective computing, natural sciences, healthcare, and HCI.

The authors evaluate 11 models (including commercial and open-source) on their benchmark. To evaluate the text generation tasks, the authors use the BARTScore metric. The authors also perform human evaluations to measure how well HEMM matches human judgments.

The main contributions of this work are:
1. Proposing a new benchmark consisting of basic skills, information flows, and use cases for evaluating multimodal models.
2. Analysing performance based on different modeling and training decisions such as model size, model architecture, pretraining objective, fine-tuning objective, training data, and modality processing

---

> ### Author Rebuttal · Authors · 2024-08-17
>
> Dear reviewer:
> Thanks for the constructive and insightful comments. We have addressed your comments below. Please feel free to follow up if you have further questions!
>
> **[one and few shot]** HEMM can easily support one-shot or few-shot prompting of the models, and we have added some results with one-shot or few-shot prompting using the InternVL, LLaVA-1.6, Qwen-VL-chat, and Gemini-1.5-flash models. We include some preliminary results and observations in **Figure 2** in the [PDF](https://openreview.net/attachment?id=ffOsqhBtqw&name=pdf) that we provided.
>
> We compare the performance of Gemini-1.5-flash from 0-shot to 5-shot, because most existing multimodal models do not support interleaved multimodal inputs to conduct few-shot multimodal prompting or do not support multiple image inputs well.
>
> We find that results on MemCap, December, and PathVQA are improved a lot with the help of few-shot examples, while few-shot examples do not bring much help for Screen2words and hurt the performance of NewYorkCartoon by a large margin. This indicates that multimodal in-context learning is not universally applicable across all tasks. It can help multimodal models learn from examples for certain tasks but can also introduce biases for others. Such phenomenon indicates that HEMM can help inspire new work on the mechanisms of multimodal few-shot prompting, with longer context lengths, and with interleave multimodal inputs.
>
> We will include the full results with one and few-shot prompting across all models supporting this feature into our revised paper. We have also included updated plots and trends of few-shot performance in the rebuttal pdf. Note: only those models that can process interleaved image-text input can be evaluated using/a few short prompts.
>
>
>
> **[link]** The link to our benchmark and live website can be found at https://anonymous.4open.science/r/hemm-save-06D5/. It includes all data loading, model training, and evaluation code. It also includes instructions for using HEMM and contributing datasets and models. HEMM will continue to be maintained and extended for the research community. We had previously included the HEMM code in the zipped supplementary material.
>
> **[new datasets]** The main contribution of HEMM is the taxonomy of basic skills, information flow, and real-world use cases that systematically cover the space of multimodal datasets. Furthermore, our standardized data loading, model training, and evaluation pipelines all significantly reduce the barrier of entry to training and evaluating multimodal models, enabling reproducible and reliable progress. To the best of our knowledge, there has not been a similarly motivated and realized benchmark for multimodal foundation models. If multimodal researchers want to benchmark their new models on medical or satellite imaging datasets, it would incur significant effort, whereas HEMM makes that extremely simple through a few lines of code.
>
>
> We would also like to note that from the call for papers in this track, “Benchmarks on new or existing datasets, as well as benchmarking tools.” and “Advanced practices in data collection and curation that are of general interest” are part of primary contributions for this track. Our paper provides standardization of existing datasets and rigorous, reproducible benchmarking tools for multimodal foundation models.

---

> > ### Comment · Area_Chair_Hw7X · 2024-08-30
> > **reviewer's follow-up to rebuttal**
> >
> > hi reviewer VBn8
> > this is a gentle ping that the authors have responded to your review.
> > does it address your concerns? if so, are you updating your score, if not, why?
> > appreciate the response!

---

### Official Review · Reviewer_J6Fw · 2024-07-24
**Nice work**

**Rating:** 7
**Confidence:** 4
**Correctness:** Yes, most of evaluations are appropriate
**Clarity:** Yes, this paper is well-written and h…

**Review:**

See below

**Strengths:**

1. This paper do have a very clear and comprehensive way to decompose different aspect of multimodal foundation models. Especially for the real-world use cases part, which might be a key contribution of this work compare with the previous one.
2. This are very thorough and massive experiement in this paper. It test tasks that span across over 30 datasets, with a wide coverage.
3. This paper also have a Human evaluation part that assess how well the benchmark aligns with human preferences, which is quite insightful.

**Additional Feedback:**

This benchmark, though comprehensive, does not propose new evaluation methods, data, or metrics, which might restrict the novelty of this paper.

**Documentation:**

Yes

**Limitations:**

Yes

**Opportunities For Improvement:**

1. The model list might be outdated. Include the latest models like InternVL, LLaVA-next, or closed-source models such as GPT-4, Gemini 1.5, etc.
2. Provide more analysis on model performance across different types of tasks. For example, GPT-4V performs better than BLIP in HCI but worse in healthcare. Explain why this is the case.

**Relation To Prior Work:**

Yes

**Summary And Contributions:**

A holistic benchmark for evaluating multimodal foundation models categorizes datasets along three dimensions: multimodal information flow, real-world use cases, and a taxonomy of 30 image-text datasets. This benchmark provides insights into how model size, pretraining data, instruction tuning, etc impact performance.

---

> ### Author Rebuttal · Authors · 2024-08-17
>
> Dear reviewer:
>
> Thanks for the constructive and insightful comments. We have addressed your comments below. Please feel free to follow up if you have further questions!
>
> **[new models]** We have added LLaVA-1.6, Qwen-VL-chat, InternVL, and Gemini-1.5-flash that have come out recently. We provided some preliminary results and observations with the latest added models in **Figure 1** in the [PDF](https://openreview.net/attachment?id=ffOsqhBtqw&name=pdf) that we provided.
>
> For the rebuttal, we present initial results on 5 representative datasets that cover the space of multimodal use cases (Memecap and NewyorkerCartoon for affective computing and multimedia, Decimer for science, PathVQA for healthcare, and Screen2Words for HCI). We will include the full results with LLaVA-1.6, Qwen-VL-chat, InternVL, and Gemini-1.5-flash in our revised paper. We have also included updated plots and trends of performance in the rebuttal pdf and will add these summary statistics to the main paper.
>
> The results show that Gemini-1.5-flash slightly outperforms other recent models like InternVL, LLaVA-1.6, and Qwen-VL-chat on tasks such as Memecap, Screen2Words, Decimer, and NewyorkerCartoon. However, it performs slightly worse than these models on the PathVQA dataset, where InternVL is strong. One standout finding is that Gemini-1.5-flash achieves significantly higher performance on Decimer, a dataset focused on chemical domain question answering. It far surpasses other models, with scores like InternVL (0.157), LLaVA-1.6 (0.062), and Qwen-VL-chat (0.09). This indicates that Gemini-1.5-flash is particularly well-trained in scientific and specialized data.
>
> Additionally, we observe that newer models generally perform better than older models like InstructBLIP, Mini-GPT-4, and Gemini-1.0 across most tasks, highlighting the overall advancement in multimodal abilities in recent models. Based on new models, HEMM shows progress in Vision-Language learning and also helps identify tasks that are more challenging for state-of-the-art models.
>
>
> **[analysis]** We have included additional analysis in our paper. For this particular result, GPT-4V’s addition of human feedback means that it often refuses to answer many questions in the healthcare and multimedia domains, including not answering certain medical prediction questions, not processing potentially offensive memes, and not answering some questions about human faces and gestures. The primary reason for this is GPT-4V's strict post-training alignment, which prioritizes safety and imposes high-level protections against potentially harmful responses. For other HCI questions on user interfaces and screenshots, we suspect that GPT-4V has additional access to a large amount of web interface images scraped from the web, or generated synthetically since improving its ability to understand user interfaces and building web/mobile agents is their big focus. To the best of our knowledge, other benchmarked multimodal models do not have such a large focus on user interface images.

---

> > ### Comment · Reviewer_J6Fw · 2024-08-29
> > **Thank you**
> >
> > Hi,
> >
> > Thank you for your reply, I will keep my score and good luck with your paper!

---

### Official Review · Reviewer_Ddcs · 2024-07-25
**A comprehensive benchmark for multimodal foundation models**

**Rating:** 6
**Confidence:** 4
**Correctness:** Yes, it's correct.
**Clarity:** Yes, the paper is well written.

**Review:**

The paper is clear. The originality is somewhat limited.

**Strengths:**

1. The paper is well-written and easy to follow.
2. The insights on model development are interesting and important.
3. The experiments are comprehensive and clear.

**Additional Feedback:**

Please check Opportunities For Improvement above.

**Documentation:**

NA

**Limitations:**

Yes, the limitations are included.

**Opportunities For Improvement:**

1. The contribution of the paper on the data side is limited. The paper only aggregates the existing datasets and identify the evaluation dimension, instead of creating new data. For dimensions where data is rare, it's more interesting, important, and helpful to collect new data to evaluate the corresponding ability.
2. The categories/dimensions of the benchmark is confusing. In section 2, the paper first identifies 3 different dimensions: basic skills, information flow, and real-world use cases. However, in section 4.2.1, information flow is categorized as the same level with different sub-skills, and the analysis on real-world use case is missing.

**Relation To Prior Work:**

Yes.

**Summary And Contributions:**

This paper proposes HEMM, a comprehensive benchmark to evaluate the capabilities of multimodal foundation models across 3 dimensions: basic skills, information flow, and real-world use cases. The benchmark aggregate 30 existing datasets to evaluate the model from different dimensions. The paper also analyse existing multimodal foundation models and give insights on model development.

---

> ### Author Rebuttal · Authors · 2024-08-17
>
> Dear reviewer:
>
> Thanks for the constructive and insightful comments. We have addressed your comments below. Please feel free to follow up if you have further questions!
>
> **[new datasets]** The main contribution of HEMM is the taxonomy of basic skills, information flow, and real-world use cases that systematically cover the space of multimodal datasets. Furthermore, our standardized data loading, model training, and evaluation pipelines all significantly reduce the barrier of entry to training and evaluating multimodal models, enabling reproducible and reliable progress. To the best of our knowledge, there has not been a similarly motivated and realized benchmark for multimodal foundation models. If multimodal researchers want to benchmark their new models on medical or satellite imaging datasets, it would incur significant effort, whereas HEMM makes that extremely simple through a few lines of code.
>
> We would also like to note that from the call for papers in this track, “Benchmarks on new or existing datasets, as well as benchmarking tools.” and “Advanced practices in data collection and curation that are of general interest” are part of primary contributions for this track. Our paper contributes novel standardization of existing datasets and rigorous, reproducible benchmarking tools for multimodal foundation models.
>
> **[categorization]** We apologize for the typo in the categorization: subsection ‘Multimodal Skills 5: Information flow’ should be a separate section with an analysis of information flow. Analysis of real-world use cases is included at the beginning of subsection 4.2.1 under ‘Overall comparisons’, where we summarized overall trends across datasets. For example, models perform better on multimedia use cases, with IRFL (0.58), NLVR (0.50), and Winoground (0.49) showing the highest scores. The lowest scores are for Healthcare, HCI, and Science use cases, such as on Decimer (0.07), iNaturalist (0.08), ENRICO (0.12), PathVQA (0.15), and Memecap (0.32).

---

> > ### Author Response · Authors · 2024-08-26
> > **follow-up to rebuttal**
> >
> > Hi reviewer, we were wondering if you had any additional concerns after our rebuttal. Please let us know and we will be happy to engage further.

---

### Author Rebuttal · Authors · 2024-08-17

We thank all reviewers for their valuable suggestions. Below, we address some common concerns raised across the reviews.

**[contribution]** The main contribution of HEMM is the taxonomy of basic skills, information flow, and real-world use cases that systematically cover the space of multimodal datasets. Furthermore, our standardized data loading, model training, and evaluation pipelines all significantly reduce the barrier of entry to training and evaluating multimodal models, enabling reproducible and reliable progress. To the best of our knowledge, there has not been a similarly motivated and realized benchmark for multimodal foundation models. If multimodal researchers want to benchmark their new models on medical or satellite imaging datasets, it would incur significant effort, whereas HEMM makes that extremely simple through a few lines of code.

**[new models]** We have added LLaVA-1.6, Qwen-VL-chat, InternVL, and Gemini-1.5-flash that have come out recently. We provided some preliminary results and observations with the latest added models in **Figure 1** in the [PDF](https://openreview.net/attachment?id=ffOsqhBtqw&name=pdf) that we provided.

For the rebuttal, we present initial results on 5 representative datasets that cover the space of multimodal use cases (Memecap and NewyorkerCartoon for affective computing and multimedia, Decimer for science, PathVQA for healthcare, and Screen2Words for HCI). We will include the full results with LLaVA-1.6, Qwen-VL-chat, InternVL, and Gemini-1.5-flash in our revised paper. We have also included updated plots and trends of performance in the rebuttal pdf and will add these summary statistics to the main paper.

The results show that Gemini-1.5-flash slightly outperforms other recent models like InternVL, LLaVA-1.6, and Qwen-VL-chat on tasks such as Memecap, Screen2Words, Decimer, and NewyorkerCartoon. However, it performs slightly worse than these models on the PathVQA dataset, where InternVL is strong. One standout finding is that Gemini-1.5-flash achieves significantly higher performance on Decimer, a dataset focused on chemical domain question answering. It far surpasses other models, with scores like InternVL (0.157), LLaVA-1.6 (0.062), and Qwen-VL-chat (0.09). This indicates that Gemini-1.5-flash is particularly well-trained in scientific and specialized data.

Additionally, we observe that newer models generally perform better than older models like InstructBLIP, Mini-GPT-4, and Gemini-1.0 across most tasks, highlighting the overall advancement in multimodal abilities in recent models. Based on new models, HEMM shows progress in Vision-Language learning and also helps identify tasks that are more challenging for state-of-the-art models.

**[one and few shot]** HEMM can easily support one-shot or few-shot prompting of the models, and we have added some results with one-shot or few-shot prompting using the InternVL, LLaVA-1.6, Qwen-VL-chat, and Gemini-1.5-flash models. We include some preliminary results and observations in **Figure 2** in the [PDF](https://openreview.net/attachment?id=ffOsqhBtqw&name=pdf) that we provided.

We compare the performance of Gemini-1.5-flash from 0-shot to 5-shot, because most existing multimodal models do not support interleaved multimodal inputs to conduct few-shot multimodal prompting or do not support multiple image inputs well.

We find that results on MemCap, December, and PathVQA are improved a lot with the help of few-shot examples, while few-shot examples do not bring much help for Screen2words and hurt the performance of NewYorkCartoon by a large margin. This indicates that multimodal in-context learning is not universally applicable across all tasks. It can help multimodal models learn from examples for certain tasks but can also introduce biases for others. Such phenomenon indicates that HEMM can help inspire new work on the mechanisms of multimodal few-shot prompting, with longer context lengths, and with interleave multimodal inputs.

We will include the full results with one and few-shot prompting across all models supporting this feature into our revised paper. We have also included updated plots and trends of few-shot performance in the rebuttal pdf. Note: only those models that can process interleaved image-text input can be evaluated using/a few short prompts.

---

### Decision · Program_Chairs · 2024-09-26

**Decision:**

Accept (Poster)

**Comment:**

Recommendation: accept poster

The authors propose HEMM, an extensive aggregate benchmark of image-text foundation models. The benchmarks is composed of 30 existing datasets that evaluate the models along different axes: basic skills, information flow, and real-world use cases.

Several reviewers noticed the lack novelty (reporting numbers of a handful of existing benchmarks), to which the authors answered that the core contribution of the work is the 3-categories taxonomy of the datasets along with an evaluation framework. I find the taxonomy quite confusing even after reading the paper in details, and the rebuttal to PdCU’s review makes it even more confusing (”At the data level, basic skills such as multimodal interactions and reasoning are inherent properties of different multimodal data distributions”) → why aren’t all skills a reflection of the training data distribution?

The authors addressed many of the weaknesses raised by the reviewers, in particular:
- lack of recent models in the comparison → adding LLaVA-1.6, Qwen-VL-chat, InternVL, and Gemini-1.5-flash
- few-shot/zero-shot results → to be completed for final paper

Overall, while the taxonomy is confusing, I agree with the reviewers on the value of the work and recommend an accept.